

# Challenges of high-fidelity air quality modeling in urban environments - PALM sensitivity study during stable conditions

Jaroslav Resler[1], Petra Bauerová[2], Michal Belda[3], Martin Bureš[1,4], Kryštof Eben[1], Vladimír Fuka[3], Jan Geletič[1], Radek Jareš[4], Jan Karel[4], Josef Keder[2], Pavel Krč[1], William Patiño[2], Jelena Radović[3,4], Hynek Řezníček[1], Matthias Sühring[5,6], Adriana Šindelářová[2], and Ondřej Vlček[2]

[1]Institute of Computer Science of the Czech Academy of Sciences, Pod Vodárenskou věží 271/2, 182 00 Prague 8, Czech Republic
[2]Czech Hydrometeorological Institue, Na Šabatce 2050/17, 143 00 Prague 12, Czech Republic
[3]Department of Atmospheric Physics, Faculty of Mathematics and Physics, Charles University Prague, V Holešovičkách 2, 180 00, Prague 8, Czech Republic
[4]ATEM – Studio of ecological models, Roztylská 1860/1, 148 00 Prague 4, Czech Republic
[5]Institute for Meteorology and Climatology, Leibniz University Hannover, Herrenhäuser Straße 2, 30419, Hannover, Germany
[6]Pecanode GmbH, Peterstraße 30, 38640, Goslar, Germany

**Correspondence:** Jaroslav Resler (resler@cs.cas.cz)

**Abstract.** The urban air quality is an important part of human well-being and its detailed and precise modeling is important for efficient urban planning. In this study the potential sources of errors in LES runs of the PALM model in stable conditions for a high-traffic residential area in Prague, Czech Republic with focus to street canyon ventilation are investigated. The evaluation of the PALM model simulations against observations obtained during a dedicated campaign revealed unrealistically

high concentrations of modeled air pollutants for a short period during a winter inversion episode. To identify potential reasons, the sensitivities of the model to changes of meteorological boundary conditions and adjustments of model parameters were tested. The model adaptations included adding the anthropogenic heat from cars, setting a bottom limit of the subgrid-scale TKE, adjusting the profiles of parameters of the Synthetic Turbulence Generator in PALM and limiting the model time step. The study confirmed the crucial role of the correct meteorological boundary conditions for realistic air quality modeling during

stable conditions. Besides this, the studied adjustments of the model parameters proved to have a significant impact in these stable conditions, resulting in a decrease of concentration overestimation in range 30–66% while exhibiting negligible influence on model results during the rest of the episode. This suggested that the inclusion or improvement of these processes in PALM is desirable despite their negligible impact in most other conditions. Moreover, the time step limitation test revealed numerical inaccuracies caused by discretization errors which occurred during such extremely stable conditions.

## 1 Introduction

The ever-increasing process of urbanization (United Nations, 2022) presses the issue of addressing urban environmental hazards, e.g. air pollution, heat waves, increased heat stress or disrupted thermal comfort (Geletič et al., 2023; Tian et al., 2024) and obliges cities across the world to move their focus towards environmental sustainability and human well-being (Gál and Kántor,



2020; Hamdi et al., 2020). The way city features influence the atmospheric flow, turbulence regime, and overall microclimate
by modifying temperature, moisture, radiation wind field, etc., is well-known and documented (Oke et al., 2017). According to
Hamdi et al. (2020), a prerequisite for understanding the impacts of climate change and urban population growth on the cities
are high-resolution microscale numerical simulations combined with urban climate observations and high-resolution urban
structure data. However, accurate representation of the complex urban environment (e.g., urban landscape and fabric) along
with the number of different scales involved in physical processes remains a complicated challenge in meteorological models
(Nazarian et al., 2023). Due to the current technological progress in climate models and the availability of computational re-
sources, scientists can employ such models for mitigation scenarios and strategy development, as well as for analyzing urban
atmospheres in large detail (Masson et al., 2020).

For this purpose, numerous numerical models with different physical bases have been developed ranging from simple urban
canopy models (Grimmond et al., 2011, 2010) to high-resolution building resolving models, e.g., 10 m and finer (Schoetter
et al., 2023). The latter class of models is based on the Computational Fluid Dynamics (CFD) methods and is further divided
into Reynolds-averaged Navier-Stokes (RANS) and Large Eddy Simulations (LES) based models (Blocken, 2015, 2018).

However, high-precision and turbulence-resolving methods in the models alone are not sufficient for them to be considered
fully reliable for urban atmosphere research, especially in the realm of air quality in the cities. To assess the accuracy of model
predictions for realistic urban meteorological conditions (e.g., air temperature and wind speed) and air pollution (e.g., NO,
$NO_2$, $O_3$, $PM_{10}$, and $PM_{2.5}$) on street-scale levels, the complete modeling chain including model inputs and configurations
must be validated against detailed measurements in a real environment. A further complication emerges from the urban surface
heterogeneity which cannot be sufficiently captured by standard meteorological stations due to their sparse distribution. Hence,
quality-controlled meteorological and air quality measurement campaigns have to be conducted to acquire additional data
for extensive model evaluation. Some examples of observational campaigns carried out in cities across the United States of
America and Europe are listed in the works of Hamdi et al. (2020) or Resler et al. (2021).

When validating air quality, an important aspect is the overall capability of the utilized model to accurately capture the
meteorological parameters and physical processes in the studied area due to the influence of wind speed (Cichowicz et al.,
2020), air temperature profiles (Wolf et al., 2014), and near-surface turbulent mixing (Wolf et al., 2020) on air pollutant
dispersion and spatio-temporal distribution. Over the years, atmospheric pollution modeling has been of interest to many
authors who use different modeling approaches (e.g., Letzel et al., 2008; Wolf et al., 2014, 2020; Jeanjean et al., 2016; van
Hooff et al., 2017; Grylls et al., 2019; Zhang et al., 2021; Weger and Heinold, 2023; Lumet et al., 2024; Wang et al., 2024;
Borna et al., 2024; Samad et al., 2024). However, model validation studies accompanied by observational campaigns focusing
on urban air quality within the street canyons are rather scarce and constructed mainly for the RANS-based models (Resler
et al., 2021).

The PALM model system (PALM; Maronga et al., 2020) utilized in this study has been proven to be of service for various
purposes owing to its well-rounded tools for atmospheric dynamics, energy balance, chemistry, and representation of urban
emission inventories within the modeled domain as shown in the validation study by Resler et al. (2021). It can employ realistic
boundary conditions given by the operational mesoscale models capturing the real meteorology in arbitrary areas and thus





providing the microscale model with the real physically consistent state of the atmosphere. PALM has been subjected to many
studies analyzing its performance from different aspects. For instance, Belda et al. (2020) tested its sensitivity to different
parameters of urban surfaces through a series of different scenarios, Salim et al. (2022) focused on differentiating between
the importance of radiative processes modeled by the radiative transfer model (RTM; Krč et al., 2021), and Radović et al.
(2023) tested its sensitivity to different mesoscale driving conditions. Lo and Ngan (2015) studied ventilation characteristics
of a simple street canyon with PALM using the tracer age approach, Lo and Ngan (2017) used PALM for investigation of the
Lagrangian residence and exposure times, and tests of the effects of lateral openings on courtyard ventilation and pollution
with PALM can be found in Gronemeier and Sühring (2019).

Tests of sensitivities of LES models under the stable boundary layer conditions are presented in some studies, e.g. Maronga
and Li (2022) tested the grid sensitivity and Couvreux et al. (2020) tested the sensitivity to the domain resolution and size,
initial profiles, subgrid-scale (SGS) parametrization, and surface roughness length and surface-flux parameterization. They
state that stable boundary layers represent a special challenge for LES models. A set of sensitivities testing the influence of
internal and external factors on the street canyon ventilation in real urban conditions during stable conditions in combination
with validation backed up by the quality-controlled observational campaign focused on air quality, to the best of our knowledge,
has not been performed yet.

Establishing a complete framework for a modeling system to be useful for routine assessment of air quality on the street
level using a micro-scale meteorological model (e.g., PALM) requires careful selection of all its components (i.e., model
configuration, numerical parameter settings, input data), and a thorough evaluation under various conditions. Our experiment
is a part of such a framework which was specifically designed to facilitate an extensive evaluation of the PALM's performance
under multiple meteorological situations for the city of Prague. It was designed as an extension and a follow-up of the earlier
evaluation experiment (Resler et al., 2021) allowing evaluation of the detailed spatiotemporal structure of air pollution in two
transportation-burdened street canyons and their surroundings (see Section 2.1). Thus, an observation campaign encompassing
the mentioned area was deliberately designed, and it took place from the spring of 2022 until the summer of 2023. The
campaign was supported by various observational tools including low-cost sensor stations for air quality monitoring, mobile
telescopic meteorological mast, microwave radiometer, and Doppler LIDAR, accompanied by the standard measurements
from professional meteorological and air quality stations at the site of interest. A summary of the observation campaign is in
Section 2.3 and its full description is in preprint (Bauerová et al., 2024). To ensure the comparability of the modeled data with
observations, a special effort was devoted to securing detailed and precise modeling inputs, particularly information about the
urban canopy, meteorological conditions, and emission sources (see Section 2.4).

In our validation simulations, we experienced unrealistic concentration patterns during some of the evaluated episodes
which could disturb the validity of the results, e.g. by deviation of the statistics based on modeled data. The most striking
event arose during the episode covering days 13–15 February 2023. This winter-time episode was characterized by a stable
synoptic situation (see Section 2.2). Under these conditions, PALM showed a strong overestimation of aerosol air pollution on
13 February 2023 during the evening traffic peak hours 16–19 (see Fig. 1).



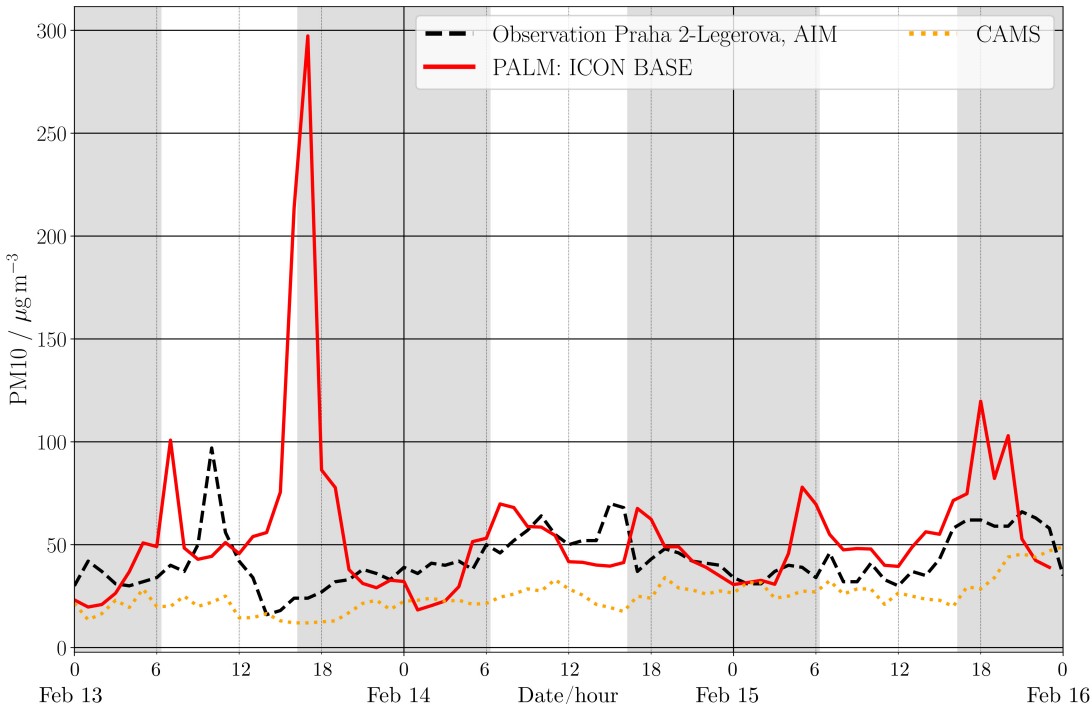

**Figure 1.** Comparison of the concentrations from the PALM simulation (red line) with observations from ALEGA AIM station (black dashed line) and mesoscale values used as BC (orange dotted line) for the entire episode 13–15 February 2023.

In this study, we target the reasons for this unrealistic behavior and we investigate the potential of different mitigation strategies. Due to high $PM_{10}$ emissions from transport inside the street canyon, this pollutant can be considered as a proxy for the
ventilation inside the street canyon, and given efforts put into the preparation of precise model inputs including emissions, our working hypothesis was that ventilation in the street canyon was underestimated during this episode. To study this phenomenon, we devised a suite of tests evaluating the sensitivities of the ventilation to processes with a potential to influence this behavior. This can provide a hint as to where to look for possible amendments for the issue in the future.

The design of the experiment, model inputs, model configuration, and utilized observations are described in Section 2, the air
quality, wind speed and turbulence comparisons are presented in Section 3. Detailed aspects of the processes and their impacts on concentration overestimation are discussed in Section 4. The manuscript closes with final conclusions in Section 5.



## 2 Experiment design

The design of the sensitivity tests is based on identified suspected reasons for the insufficient modeled ventilation in the street canyon during the studied episode. The experiment includes tests of the model driving meteorology and internal model processes and configurations; their description is given in detail in Section 2.5.

### 2.1 Study area

The study area, further called the 'model domain' or 'domain', is located in the central part of Prague (Fig. 2), the capital of the Czech Republic. The evaluation was set up to cover the so-called 'Legerova hot-spot', an area that faces various urbanistic challenges, especially connected to the very intensive traffic. This hot spot has been the focus of the Czech Hydrometeorological Institute, city planners and officials for many years. The area is centered around the Legerova and Sokolská streets which are parts of the Prague Magistrála, the North-South arterial road running through the Prague center. This road constructed in the 1970s is one of the major concerns for various reasons. One of the main problems is high traffic loads exceeding 35,000 cars per day in each direction and causing poor air quality. The second problem is the dominant building structure: the whole domain is mainly formed by compact mid-rise buildings (LCZ 2; Stewart and Oke, 2012) with small fractions of open midrise (LCZ 5). In the south part of the modeled domain, there is a deep valley with a park (LCZ B), which is bridged by the Prague Magistrála. These aspects make this area an ideal test bed for model experiments.

### 2.2 Study episode

For this analysis, a winter-time stable situation between 13 and 15 February 2023 was selected. For convenience, a climatological summary for Prague is provided in Appendix A1. Synoptically, the period was characterized by a slowly moving high-pressure system over western and central Europe (Fig. S01–S04 in Supplements). Warm air advection was observed at the 850 hPa level (Fig. S05–S07) strengthening the prevalent capping inversion (Fig. S09). The situation represents a typical strong winter-time inversion with cloudy conditions (cloud cover of 7 or 8 oktas) during the daytime.

Meteorological station Praha 2-Karlov (P1PKAR01, WMO ID 11519), located in the area of interest several hundreds of meters from Legerova street (see Fig. S13), was used as a reference. The air temperature in the Prague city center varied between -1.0 °C and 9.2 °C, the relative humidity was 45–81%, and the wind speed was lower than 10 ms$^{-1}$. On 13 February, the sky was broken to overcast with a shallow layer of low clouds. During the evening hours (near 18h UTC) there was a temporary cloud cover decay (to 1/8) but due to the deepening and lowering of the inversion a low cloud layer developed during the night and remained present till the end of the episode.

The temperature profiles and heights of the inversion layer forming during the evening can be identified in Fig. 3 where the outputs from the microwave radiometer (MWR) in Praha 2-Karlov are plotted for the whole 72 h episode. The well-mixed stratocumulus-topped convective boundary layer was present on 13 February during the day. During the night, the inversion deepened and moved lower, close to the height of 1000 m, according to the radio soundings at Praha-Libuš (P1PLIB01, WMO ID 11520), located 7 km to the south of Karlov station (for location see Fig. S13 and for more information Fig. S10 in



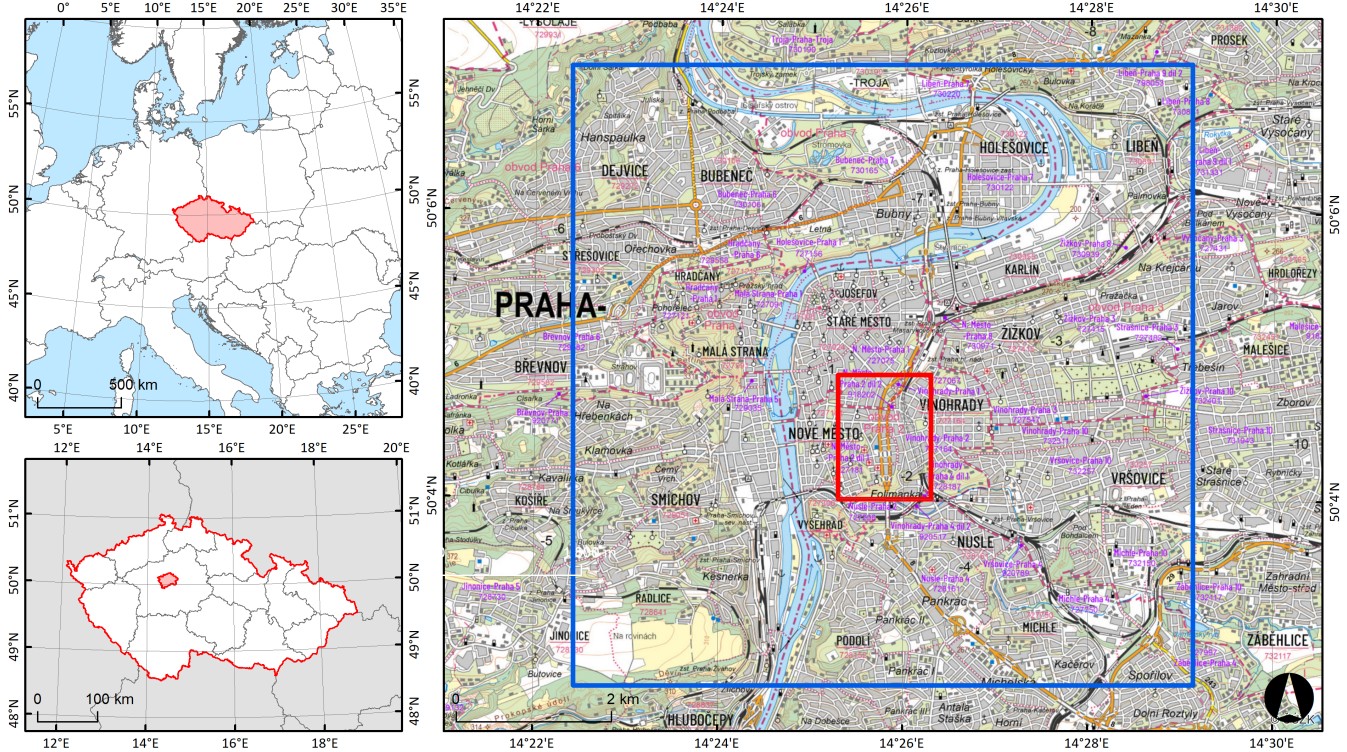

**Figure 2.** Location of the Czech Republic in Europe (top left) and the city of Prague in Czech Republic (bottom left). The map on the right represents the parent (blue rectangle) and child (red rectangle) domain including 'hot-spot Legerova'. The vector geodata for the maps on the left were provided by ESRI (Environmental Systems Research Institute, Europe NUTS 0 Boundaries), topographical map on the right map is served through the web map service of the Czech Office for Surveying, Mapping and Cadastre – ČÚZK.

Supplements). In the subsequent days of the episode, the wind speeds near the surface were very low, the sky was overcast or
broken and the deep inversion was moving even lower and on 15 February, it is visible in the microwave radiometer profiles.

The wind in the evening of 13 February is significant due to the modeled concentration peak. The wind direction was changing with height from the weak south-western flow near the surface to the northern flow in higher levels (see Fig. S08, S09, or Fig. 4 for the LIDAR observations). After the sunset (16:17 UTC), the wind accelerated up to $10\ \mathrm{ms^{-1}}$ and formed a low-level jet near the inversion forming at the top of the boundary layer around the height of 300 m (Fig. S09).

## 2.3 Observational data

Drawing from the experience of previous analyses done by Resler et al. (2017, 2021), a targeted observational campaign was set up for the PALM model performance evaluation. It includes almost one year (from 30 May 2022 to 28 March 2023) of air quality and meteorological observations. The concentration of $NO_2$, $O_3$, $PM_{10}$ and $PM_{2.5}$ were measured by 20 combined low-cost sensors placed in different sites and different heights above ground. Observed values from sensors were subsequently



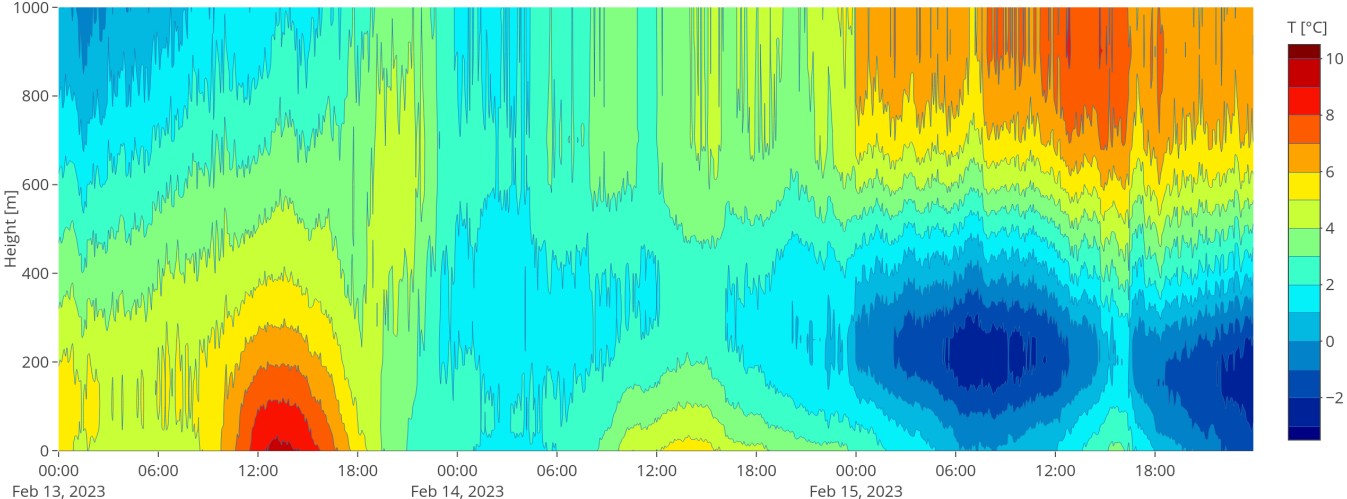

**Figure 3.** Vertical profile of temperature T in [°C] measured by MWR for days 13–15 February 2023. The weather change and forming of the inversion during the 13. February evening can be identified at 18:00. The height variable denotes the height above the radiometer located at altitude 261 m a.s.l.

corrected using the sensor co-measurement with the reference station Praha 4-Libuš (ALIBA) of the Automatic Air Quality Monitoring system (AIM) at the beginning and the end of the campaign. Additionally, one professional meteorological station Praha 2-Karlov and one reference traffic AIM station Praha 2-Legerova (ALEGA) are located within the child modeling domain. To gain high spatial and temporal resolution meteorological data for the studied area, the supplementary measurements were established; it consisted of i) mobile telescopic meteorological mast for measuring temperature, relative humidity, wind velocity and direction and air pressure; ii) MTP-5-He microwave radiometer for the temperature vertical profile; iii) StreamLine XR Doppler LIDAR for the wind vertical profile. The location of the measurements in a map is shown in Figure S13 and additional information about the measurement stations is in Table S11 in Supplements. A detailed description of the observational campaign is available in (Bauerová et al., 2024).

### 2.4 Modeling setup

The PALM model system release 23.04 was used in this experiment. The particular code used was derived from the PALM gitlab master development branch from 24 May 2023 with added fixes and adaptations needed for simulations in this experiment. The model was configured with the LES core and it solves non-hydrostatic, filtered, Boussinesq-approximated, incompressible Navier-Stokes equations. The subgrid stress tensor is modeled by the Deardorff (1980) 1.5-order closure involving Moeng and Wyngaard (1988) and Saiki et al. (2000) modifications. For spatial and temporal discretization, the upwind-biased 5th-order differencing scheme Wicker and Skamarock (2002), and the 3rd-order Runge–Kutta time-stepping scheme Williamson (1980) were used, respectively. The solution of the pressure-Poisson equation in the projection step was configured to utilize the multi-grid scheme solver (e.g., Maronga et al., 2020). In addition to the core system, the PALM modules developed for studying the



urban boundary layer were employed. They include e.g. the land surface model (LSM; Gehrke et al., 2021), the building

surface model (BSM; Resler et al., 2017; Maronga et al., 2020), the radiative transfer model and the plant canopy model (RTM

and PCM; Krč et al., 2021), online nesting Hellsten et al. (2021), mesoscale nesting Kadasch et al. (2021), and the chemical

transport model (Khan et al., 2021).

This study concentrates on assessing concentrations of $PM_{10}$. This pollutant was selected as it does not undergo significant

changes in the usual timeframe of the passing of the air through the PALM domain (unlike the ozone-chemistry-related species

NO and $NO_2$) and thus it can be modeled and evaluated as a passive tracer. This allows us to use it as a proxy for the assessment

of the ventilation inside the street canyon.

To properly simulate all mixing layer scales, PALM was configured in two nested domains (see Fig. 2). The parent domain

had an extent of 8,000 m × 8,000 m with a horizontal resolution of 10 m and a height of 2,750 m. The vertical resolution

of the parent domain was 10 m and the vertical stretching was applied from level 400 m with a stretching factor of 1.08 and

an upper limit of 20 m. The nested child domain had an extent of 1,200 m × 1,600 m, a horizontal and vertical resolution of

2 m, and a height of 320 m with no vertical stretching. The modeling setup was configured with the full 3D geometry which

allows to model objects such as bridges and multilevel crossroads. The input emission was provided in the form of the recently

developed PALM general volume sources which allowed to build the correct vertical structure of the emission flow.

The simulation was initialized in two steps. First, temperature of soil, building walls, and pavement layers was simulated

using the PALM built-in spin-up feature. During this phase, the dynamic part of the model was switched off and only the energy-

balance related modules (RTM, BSM, and LSM) were employed (see Maronga et al., 2020). Next, six hours of initialization

simulation followed with the full model processes. Both initialization steps were excluded from the evaluation.

### 2.4.1 Static input data

For detailed modeling of micro-scale processes utilized in the PALM model system, high-fidelity spatial data are needed. Due

to the influence of surface thermal heating on airflow by buoyancy, processes of surface energy balance must be accurately rep-

resented. The correct modeling of the energy balance equations in BSM, LSM and RTM requires proper estimates of numerous

input parameters characterizing physical properties of horizontal (roads, pavements, etc.) and vertical surfaces (walls) as they

have high influence on radiative and heat exchanges (Belda et al., 2020). Surface parameters needed are e.g., albedo, emissivity,

thermal conductivity, roughness length, characteristics of the skin layer and subsurface layers (thermal capacity and volumetric

thermal conductivity). An updated GIS database, previously introduced by Resler et al. (2021), was used for the present study.

This database includes information on wall, ground, roof materials and finishes as well as roughness and window fractions,

which were used to estimate surface and material properties. Each surface was described in terms of material category, albedo,

and emissivity. Urban greenery, e.g. individual trees and shrubs, was described by its position, trunk height and diameter, tree

height and crown diameter, crown shape, and tree type. The leaf area density (LAD) based on tree types was then calculated

according to the irradiation/shading profile of the particular part of the tree crown (distance from the border of the crown). For

deciduous trees in winter, only branches and trunk were modeled and they were roughly estimated as 10% of the LAD of the



summer tree. However, Legerova and Sokolská streets which are in the center of interest contain almost no trees. All these data were processed into the PALM Input Data Standard (PIDS) format.

### 2.4.2 Initial and lateral boundary data

The meteorological initial and lateral boundary conditions (IBC) for the base case PALM simulation were provided by the
operational simulation of the numerical weather prediction model ICON run at the German national meteorological service Deutscher Wetterdienst (Reinert et al., 2020). The data were taken from the ICON-D2 regional simulation with a horizontal resolution of approximately 2.2 km. Due to the unavailability of certain required variables within the assimilation run archives, the data from the main forecast run were used with the shortest prediction horizon taken for every simulated hour. The selection of the ICON model resulted from a comparison of three meteorological models (ICON, WRF, and ALADIN) with sounding
observations performed operationally by the Czech Hydrometeorological Institute (CHMI) at the nearest sounding station Praha 2-Libuš for multiple episodes of the year (see Tables S01–S09 in Supplements). The other models (ALADIN and WRF) were used in this study for the tests of sensitivity to the meteorological boundary conditions (BC). The ALADIN model data were taken from the standard operational forecast provided by CHMI ((Termonia et al., 2018); adaptation for the Czech Republic by Brožková et al. (2019)); the assimilation cycle starting at 00, 06, 12, and 18 UTC with 6 h prediction horizon. The
WRF simulations were driven by ERA5 reanalysis (Hersbach et al., 2023) and the model configuration was selected based on the results of the comparison of more different model setups (Radović et al., 2023).

The mesoscale model outputs used as the boundary conditions differed in capturing the correct wind speed and direction during the peak period (see Fig. 4 and for whole day Fig. S11 in Supplements). Before the sunset, all models underestimated the velocities in the boundary layer (at least below 700 m) but they differed in terms of wind directions. During the following hours,
all models started to increase the wind speed near the ground. The quickest was ALADIN, which started to accelerate the wind almost immediately after sunset (16–17h). WRF gave the large wind speed later (between 17–18h), but ICON underpredicted the velocities in the domain for the entire peak time. ICON also gave the most unrealistic wind directions in the first two hours of the concentration peak episode (16–17h). The statistical comparison of all three mesoscale models with the LIDAR and MWR observations for the whole day is shown in the Table S10 in Supplements. The comparison of the profiles of
the modeled potential temperature with observations from MWR for the peak period is in Fig. 5, the complete set of the profiles for the whole day is in Fig. S12. The profiles show that the mesoscale models were, in general, not able to completely capture the statically-stable temperature gradient (inversion) that was forming below the strong inversion after the sunset. They form a strongly stable boundary layer and a significantly less-stable residual layer above. That is unrealistic according to the measurement. ICON and WRF predicted lower surface temperatures around sunset (16–17h) and the effect occurred earlier for
them.

The initial and boundary conditions for air pollutants were the result of analysis combining data from the AIM stations with CAMS ENSEMBLE model (CAMS, 2020): the median value of measurements from AIM background stations inside and in the surroundings of the domain (i.e. stations Praha 5-Stodůlky (ASTOA), Praha 8-Kobylisy (AKOBA), Praha 4-Chodov (ACHOA), Praha 4-Libuš (ALIBA); see Fig. S13 and Table S11 in Supplements) was complemented by the CAMS ENSEMBLE model





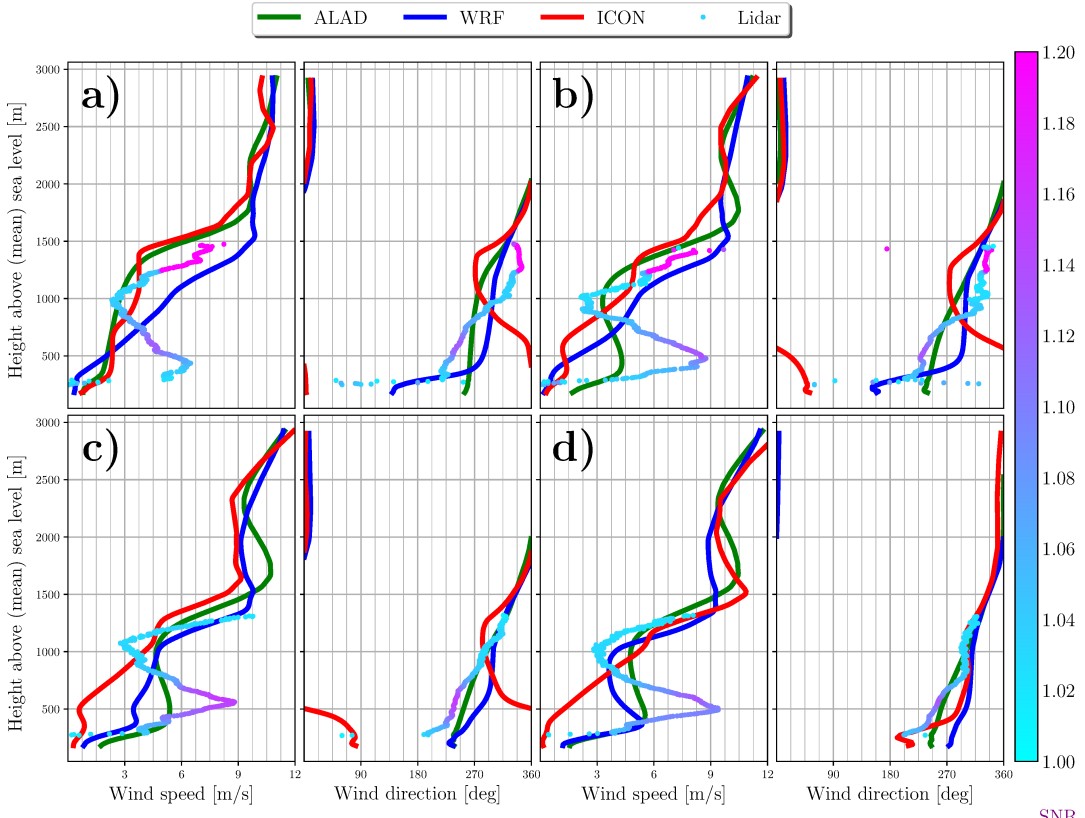

**Figure 4.** (a-d): Comparison of the wind speed and wind direction profiles modeled by mesoscale models ICON (red), ALADIN (green) and WRF (blue) used as PALM model BC with profiles from LIDAR observations. The color of the LIDAR observation (dots) shows the LIDAR signal/noise ratio and the uncertainty of the observation. The displayed hours cover the concentration peak of the 13. February: 16 (a), 17 (b), 18 (c), and 19 (d) UTC. Values represent hourly averages for the hour ending in the given time.

based vertical profiles scaled to observed near-surface value. Therefore, for a given hour, boundary conditions were identical for all the boundaries.

The dynamic driver was prepared by using the recently developed tool PALM-Meteo (see section Code and data availability), which is a part of the PALM supporting tools. This tool can take inputs from different mesoscale meteorological and air quality models, transform them to the values needed for PALM, and store them in the format of the PALM dynamic driver according to the PIDS.

### 2.4.3 Emissions

As this study focuses on comparing air quality values, the preparation of the precise emission data was substantial. The data were processed mainly from datasets collected by CHMI and by the Municipality of Prague and its organizations, data obtained



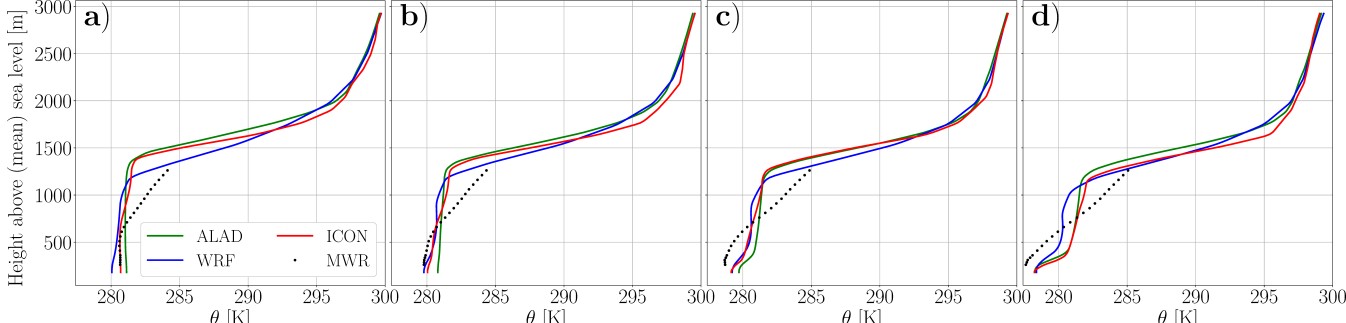

**Figure 5.** (a-d): Comparison of the potential temperature (Θ) profiles modeled by mesoscale models ICON (red), ALADIN (green) and WRF (blue) used as PALM model BC with profiles from MWR observations (dots). The displayed hours cover the concentration peak of the 13. February: 16 (a), 17 (b), 18 (c), and 19 (d) UTC. Values represent hourly averages for the hour ending in the given time.

by the researcher (ATEM) while providing expert studies in the past, and results of previous research projects. The input data of the used emission sources can be divided into two basic groups: emission from local heating and transport sources.

Emissions for local heating were determined by calculations based on data from CHMI and the Czech Statistical Office. Emissions from the transport sources were modeled using the MEFA transportation emission model (MEFA, 2013) which is recommended for the use in the Czech Republic by the Ministry of Environment of the Czech Republic. The model takes into account factors such as road gradient, the number of vehicles on the road, the flow of traffic, the composition of car types, and the emission characteristics of the individual car types. The emission calculation is based on data from the traffic census provided by the Prague Technical Administration of Roads (TSK Praha) and on data from the census of the composition of the transportation fleet in Prague built in the MEFA emission model. The data are based on regular surveys of the fleet composition carried out in Prague (Karel et al., 2021). The dust resuspension was computed according to the methodology published by the Ministry of Environment (Karel et al., 2015). This methodology is based on US EPA methodology AP-42 (EPA, 2011) and was adjusted for the conditions of the Czech Republic. For the garages and parking lots, the results of the project TH03030496 (Karel et al., 2020) were used. For the bus stations, publicly available data about transportation were gathered from the Prague Public Transit Company (DPP).

The disaggregation of the annual emissions into hourly intervals was then performed according to the type of source. For combustion sources, distribution of emissions to days was done according to natural gas supply profiles for category DOM4 (OTE, 2024) and complemented by daily profiles for SNAP 2 (Van der Gon et al., 2011). For transport sources, the census data from TSK Praha was utilized for all streets where it was available. For Legerova and Sokolská streets, hourly traffic intensity data were obtained and used directly for the selected episodes. For streets that were not covered by regular traffic surveys, the spatial and temporal distribution of the traffic intensities were based on analysis and evaluation of the relevant studies for the particular area, e.g., urban planning studies, Environmental Impact Assessment (EIA, 2007), etc. Then they were combined with information like street type, location, traffic regime, and pavement type. This approach allowed us to specify the distribution of the transportation intensities on smaller streets. For the detailed modeling of emissions from rail





transport (diesel locomotives), the data of train rides were obtained from the Railway Administration (SŽ) and emission factors from the EMEP/EEA Air Pollutant Emission Inventory Guidebook 2019 (EEA, 2019) were used. Emissions from river ships were obtained from the CHMI national database and spatially distributed to the area of the river.

Spatial transformation of the line and point emission into the corresponding areas was done with the utilization of surrogates representing corresponding areas (e.g. areas of the street traffic lines and parking places for traffic emission and areas of the building roofs for local heating sources). This not only ensured the reasonable spatial distribution of the emissions in the street canyon but also decreased the gradients of the emission field and, through this, the proneness of the model to numerical inaccuracy (Ardeshiri et al., 2020). The processing of the emission sources into hourly emission flows in PIDS was done in the

emission model FUME recently extended for processing of the PALM emissions (Belda et al., 2024).

## 2.5   Scenario modeling setup

To assess the working hypothesis and possible reasons for the behavior of the model during the evening hours of 13 February 2023, a range of scenarios of meteorological inputs and individual PALM processes were simulated. The list of the sensitivity tests is in Table 1. A detailed description follows in Sections 2.5.1, 2.5.2, and 2.5.3.

| Group | Scenario | Description | Parameter setting |
|-------|----------|-------------|-------------------|
| I. Meteorological scenarios - initial and boundary conditions source model | | | |
| | PALM-ICON | Driving model ICON (base case) | |
| | PALM-ALAD | Driving model ALADIN | |
| | PALM-WRF | Driving model WRF-ERA5 | |
| II. PALM processes and configuration adjustments | | | |
| | BASE | Standard model configuration (base case) | |
| | HEAT | Added anthropogenic heat from cars | |
| | SGS | Bottom limit of subgrid-scale energy | sgsmin = 0.02 m$^2$s$^{-2}$ |
| | STG | Adapted profiles for synthetic turbulent generator | |
| | DTMAX | Upper limit of model time step | dtmax = 0.2s |

**Table 1.** List of performed sensitivity tests

### 270   2.5.1   Meteorological scenarios

The meteorology from the ICON model used for the creation of the PALM boundary conditions was selected as the best input based on the evaluation of multiple episodes of the year against sounding observations on the meteorological station Praha 4-Libuš. The detailed assessment of the model results against LIDAR and MWR observations inside the modeled domain showed that ICON is not superior for this area and episode in the wind speed and is the worst one in the wind direction (see Table S10 in

Supplements). To assess the PALM response to differences between alternative meteorology inputs, simulations of the PALM





model driven by ICON (base case scenario PALM-ICON), ALADIN (scenario PALM-ALAD), and WRF-ERA5 (scenario PALM-WRF) simulations were tested for all three days 13–15 Ferbruary 2023.

### 2.5.2 PALM model processes and configurations

Sources of uncertainty were also identified in the PALM model processes and their settings. In addition to the base PALM
configuration used for the validation (scenario BASE), four additional model configurations (scenarios HEAT, SGS, STG, and DTMAX) were simulated. To keep the experiment computationally feasible, the tests of model processes focused on the first day of the studied episode where the unrealistic behavior of the model occurred.

**HEAT: Anthropogenic heat from cars**

The current version of the PALM model does include the possibility to model the anthropogenic heat and turbulence induced
by moving cars. The influence of the heat from cars on simulated air temperature proved to be minimal during a heat wave episode (Juruš et al., 2016). On the other hand, the situation during stable winter conditions can be different and this scenario was included to assess the influence of this heat on the ventilation of the street canyon. As a proxy value of the heat flow, the model emission flow of $PM_{10}$ from transportation was used and transformed into the corresponding heat flow. The calculation used is provided in Supplements in Section F.

**SGS: Bottom limit of subgrid-scale energy**

This sensitivity test was inspired by the usual practice in mesoscale chemical transport models (CTM) to set the bottom limit of the eddy diffusivity; e.g. the model CMAQ limits this value to a minimum of $1.0\,\mathrm{m^2s^{-2}}$ (see e.g. CMAQ, 2019). This practice is based on the experience that the diffusivity calculated from meteorology provided by mesoscale meteorological models is underestimated during low wind stable situations; see e.g. Byun et al. (1999) for description and Makar et al. (2014) for tests of
the influence of different settings of this limit. In the LES model PALM, the diffusivity is internally calculated by the subgrid model and depends on the modeled subgrid-scale turbulent kinetic energy (SGS-TKE). Our working hypothesis assumes that the SGS-TKE and consequently the diffusivity can be underestimated in PALM in conditions of the studied episode and this scenario applies the bottom limit for SGS-TKE. The value of the limit $0.02\,\mathrm{m^2s^{-2}}$ was selected as the maximal value of SGS-TKE modeled in the child domain at the time of the concentration peak. It also corresponds to values of SGS-TKE in the street
canyon before and after the peak in the BASE scenario (see Fig. 7 in Section 3).

**STG: Adapted profiles for synthetic turbulent generator**

Another potential reason for the insufficient street canyon ventilation during this episode could be the turbulence coming from the boundary of the outer domain. As the boundary conditions provided by a mesoscale model to PALM contain only the wind components and no information about the turbulence, the PALM's internal Synthetic Turbulent Generator (STG) module
was utilized in all PALM configurations. This generator was used on the border of the parent domain to add turbulence to the



mesoscale-provided wind flow to generate a turbulent inflow condition. STG uses the approach proposed by Xie and Castro (2008) and Kim et al. (2013); see Kadasch et al. (2021) for more details. The parameterization used in the STG is based on a maritime boundary layer and has not been extensively tested under stable conditions.

To test the hypothesis that insufficient turbulence provided by STG influences the street canyon ventilation, this scenario adjusts the conditions of PALM STG by supplying vertical profiles of values of Reynolds stress tensor and turbulent length- and time-scales for every simulation hour. These profiles add $1 \text{ m}^2\text{s}^{-2}$ to the diagonal terms of Reynolds stress tensor inside the boundary layer which is almost twice as much as the original setting $0.5 \text{ m}^2\text{s}^{-2}$ in the maximum which was given by the original STG profile for the shear-driven boundary layers. This setting allows STG to generate higher TKE on the boundaries.

**DTMAX: Upper limit of model time step**

The aim of this scenario was to detect possible inaccuracies in the numerical solution of the differential equations in PALM. PALM calculates the base integration time step as the maximal value satisfying the Courant–Friedrichs–Lewy (CFL) criterion for the Runge-Kutta scheme for the differential equations of air advection, diffusion, energy balance, and precipitation. This time step is also used for the transport of the chemical species implemented as a passive scalar for diffusion and advection. In our case, the CFL criterion applied for advection guarantees a time step exceeding 0.7s on 13 February 2023 at hours 17–
20. This large time step, together with relatively high concentration gradients around the traffic emission sources during the evening traffic peak, can make the upwind-biased 5th-order differencing scheme prone to dispersion and dissipation errors and could cause an inaccurate representation of the near-source dispersion. Moreover, this advection scheme is only conditionally conservative if the divergence is negligible. This is not necessarily met in our simulations with noticeable terrain structures like buildings. In this case, the advection term can become a source or sink of a scalar. To mitigate both these issues, various
measures can be taken, one of them is a smaller model time step. To test this hypothesis, the time step maximum was limited to 0.2s for the whole simulation.

### 2.5.3 Combined scenarios

To obtain the full set of the combinations, we also added scenarios combining meteorological scenarios PALM-ALAD and PALM-WRF with all scenarios modifying PALM processes except DTMAX. To keep the experiment computationally feasible,
these combined scenarios were calculated only for the most relevant hours 15–24 and they started from the simulation of the respective meteorological scenario in hour 14. The scenario DTMAX was omitted here as it represents a purely technical test of the accuracy of the numerical solution in the PALM model and its combinations with other BCs would bring no new substantial information and waste computational resources.





## 3 Results

### 3.1 Effects of meteorological boundary conditions

Fig. 6 shows a comparison of the modeled and observed $PM_{10}$ street-level concentrations in the AIM station ALEGA for PALM simulations with different driving meteorology. All model results presented in this and the following sections come from the fine resolution child domain if not stated otherwise. The graph suggests that boundary conditions could play a crucial role in the explanation of the differences between model simulation and observations. The over- or under-estimation of individual model scenarios differs and changes in time. The strong overestimation of the PALM-ICON on 13 February 2023 at hours 16–19 UTC (further called "peak period") is only partially seen in PALM-WRF while PALM-ALAD agrees well with observations at those hours. Besides this disagreement, PALM-ICON underestimates $PM_{10}$ during the night on 14 February (hours 00–06) similarly to other models and it follows observations well during the rest of the episode. The maximum concentration of the PALM-ICON is almost 300 $\mu gm^{-3}$ at hour 17 which overestimates the observation about 10 times. The overestimation of the PALM-WRF model against observation is lower with the maximum of about 105 $\mu gm^{-3}$ at hour 16 which is more than 4 times higher than the observation at the same time. Moreover, this overestimation is limited only to hours 16–17. In the rest of the episode, PALM-WRF overestimates the observation during the evening of 14 February and most of the day on 15 February The PALM-ALADIN model captures the observations almost perfectly during the afternoon and evening of 13 February and its results during the rest of the episode are similar to the PALM-ICON basecase model with slight underestimation in the morning and overestimation in the evening of 14 February.

Besides the main peak in the PALM-ICON concentrations of $PM_{10}$, we can see a small peak in the morning at hours 08–09. It could lead to the conclusion that PALM-ICON was the only model able to catch the similar small morning peak in observations at 10 a.m., despite the 2 hours time shift. But as a similar peak does not occur in observations of $PM_{10}$ in the nearby sensor stations S14 and S15 (see Fig. S15 in Supplements) and moreover, it does not occur in the observations of $PM_{2.5}$ on the ALEGA station (see Fig. S16), we suppose that this small peak in observations was an effect of a non-standard local emission of $PM_{10}$ nearby ALEGA station (e.g. dust from building operations or street cleaning) and is not connected with the PALM-ICON model peak in hours 08–09.

The statistics shown in Tables 2 and 3 demonstrate that the studied short unrealistic peak in PALM-ICON concentrations significantly disturbs the mean values and standard deviation of the whole three-day episode. Omitting these hours from the statistics brings the performance of PALM-ICON to a similar level as PALM-ALAD.

Additional information about the situation with respect to vertical profiles of the wind speed and wind direction are given in Fig. 7 for the peak period as well as in Fig. S17 in Supplements for the entire day. Wind speed is strongly underestimated in the PALM-ICON model, moreover, PALM-ICON gives an incorrect wind direction, which is almost opposite to the observation between hours 16–17. Comparison with the profiles of the BC provided by mesoscale models (Fig. 4) shows that PALM generally follows the BC profiles with some minor improvements, e.g. wind direction of PALM-ICON in hours 16 and 18. Further, it becomes obvious from Fig. 5 that the potential temperature inflow profiles obtained from the mesoscale models





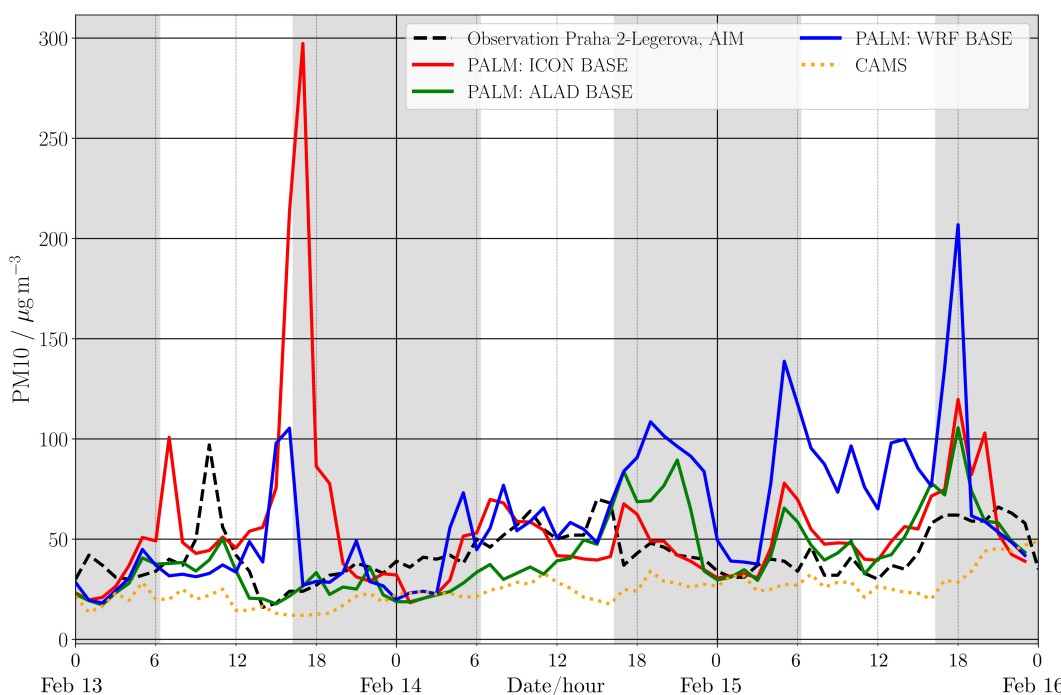

**Figure 6.** : Comparison of the PALM modeled and observed concentrations of PM$_{10}$ on the AIM station ALEGA. Observations are represented by the red dotted line and the scenarios are shown by solid lines: base case PALM-ICON (red), PALM-ALAD (green) and PALM-WRF (blue). The orange dotted line (CAMS) represents the analysis of the observations and CAMS model used as the BC.

| Values | OBS | PALM-ICON | PALM-ALAD | PALM-WRF |
|---|---|---|---|---|
| Mean (entire episode) | 42.8 | 55.3 | 41.6 | 60.9 |
| Median (entire episode) | 40.0 | 48.0 | 37.5 | 53.8 |
| Mean (without peak) | 44.8 | 48.0 | 43.6 | 61.7 |
| Median(without peak) | 41.0 | 45.7 | 39.1 | 55.0 |

**Table 2.** The mean and median of the observations on ALEGA AIM station and modeled values of scenarios PALM-ICON, PALM-ALAD, and PALM-WRF for the entire episode 13–15 February 2023 and for the episode without the concentration peak hours 13–19

deviate significantly from the observation. More precisely, the near-surface stratification (up to z = 300 m) is overestimated compared to the observation, which suggests less turbulent mixing and altered wind profiles.





| Errors | PALM-ICON | PALM-ALAD | PALM-WRF |
|---|---|---|---|
| Mean (=bias) (entire episode) | 8.0 | −1.3 | 17.5 |
| Mean (without peak) | 3.2 | −1.2 | 16.9 |
| Standard Error (entire episode) | 3.0 | 1.6 | 3.0 |
| Standard Error (without peak) | 2.4 | 2.4 | 4.4 |
| Median (entire episode) | 3.0 | −2.4 | 6.3 |
| Median (without peak) | 1.7 | −4.1 | 6.3 |
| Standard Deviation (entire episode) | 34.6 | 18.5 | 35.1 |
| Standard Deviation (without peak) | 19.6 | 19.0 | 35.2 |

**Table 3.** The statistics (mean, standard error, median, standard deviation) for errors of the model against observations on ALEGA AIM station and modeled values for scenarios PALM-ICON, PALM-ALAD, and PALM-WRF for the entire episode 13–15 February 2023 and for the episode without the concentration peak hours 13–19.

Comparison of the modeled wind speed, resolved TKE, and subgrid-scale TKE in the street canyon for scenarios PALM-ICON, PALM-ALAD, and PALM-WRF (Fig. 8) shows that the overestimation of the concentrations in PALM-ICON in hours 16–19 UTC and PALM-WRF in hours 15–16 UTC agrees with the times with very low modeled wind speed and TKE in these models. The ratio of the PALM-ALAD vs. PALM-ICON modeled values of wind speed reaches maxima over 6, resolved TKE about 200, and subgrid-scale TKE about 40.

## 3.2 Effects of the PALM processes

Comparison of the observed concentration of $PM_{10}$ on station ALEGA with the PALM-ICON model results for scenarios BASE, HEAT, SGS, STG, and DTMAX (see Fig. 9 (a)) shows how much the modeled street canyon ventilation is influenced by adaptations of the particular scenarios. It shows that all scenarios differ very little from the base simulation except for the time of the peak period and partly during another small peak in the morning (hours 07–08). In hours of the main peak, the concentrations in all scenarios are significantly lower than the BASE simulation but they still overestimate the observation. The most noticeable change can be observed for the scenario HEAT (heat from cars) with a decrease of concentrations from about 300 to 120 $\mathrm{\mu g\,m^{-3}}$ (decrease to 40%) and the scenario DTMAX (limited model time step) which decreases concentrations approx. to 155 $\mathrm{\mu g\,m^{-3}}$ (decrease approx. to 50%). Even so the maximum ratio of the modeled against observed values still remains about almost 5 for HEAT and more than 6 for DTMAX.

The comparison of the modeled time series of wind speed in the area of ALEGA station (Fig. 9 (b)) shows that street canyon wind speed is influenced very little by the adaptation of the scenarios. It is slightly and unsystematically increased in hours 12–18 UTC. A similar situation can be seen in the comparison of the vertical profiles of the modeled wind speed and wind direction with observations from LIDAR in Fig. S18 in Supplements. Comparison of the modeled time series of the resolved and subgrid-scale TKE (Fig. 9 (c) and (d)) shows that the resolved TKE does not change significantly except for a slight increase in the scenario STG in most hours and that the subgrid-scale turbulence also does not change too much except for the



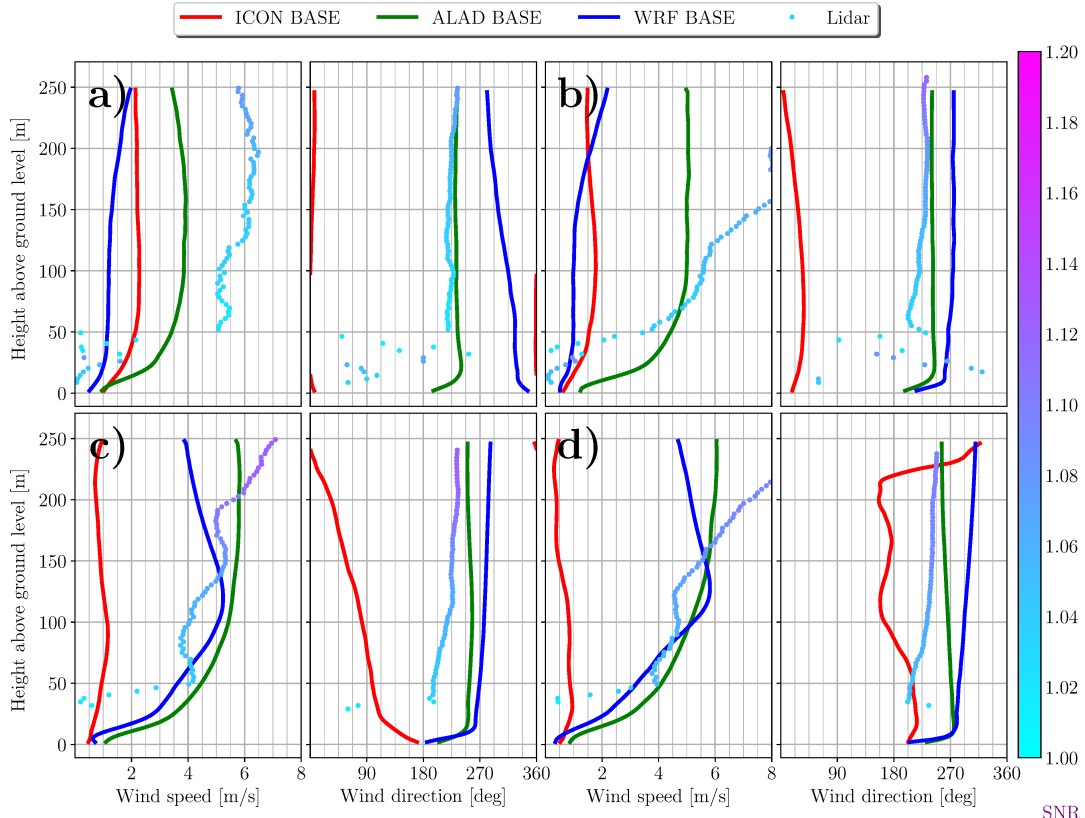

**Figure 7.** (a-d): Comparison of the PALM modeled wind speed and direction profiles from scenarios PALM-ICON (red), PALM-ALAD (green), and PALM-WRF (blue) with profiles observed on the LIDAR station (dots). The color of the LIDAR observation dotted line shows the LIDAR signal/noise ratio and the uncertainty of the observation. The displayed hours 16 (a), 17 (b), 18 (c), and 19 (d) cover the concentration peak. Values represent hourly averages for the hour ending in the given time.

increase of its values in the SGS scenario to the values slightly higher than $0.02\ \mathrm{m^2s^{-2}}$ which is the lower limit SGS-TKE set in this scenario.

### 3.3 Effects of PALM processes with alternative meteorologies

To get a complete overview of the sensitivities, the sensitivity tests to model parameters were applied also to the meteorological scenarios PALM-ALAD and PALM-WRF. From computational feasibility reasons, these sensitivities were calculated for the most interesting hours 14–24. Results (see Fig. 10) show very small differences between scenarios based on the PALM-ALAD model in all hours and PALM-WRF model in hours 17–24. In hours 15–16 where the PALM-WRF model also overestimates observations, the scenario HEAT decreases concentration from 105 to 65 µgm$^{-3}$ (decrease to approx. 62%) while the scenarios SGS and STG do not significantly change the concentration inside the street canyon. Also, there are no significant differences of the modeled wind profiles between these scenarios except slight modification of the wind speed profile for STG scenarios with

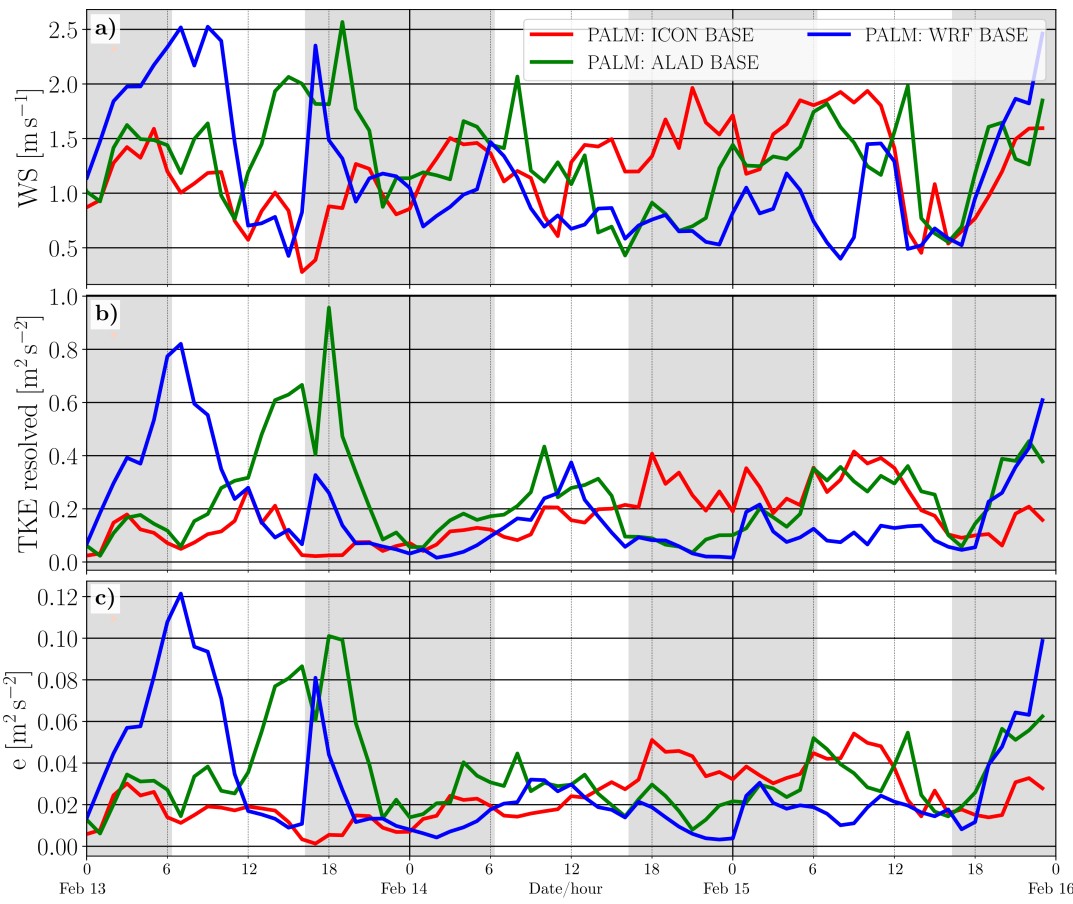

**Figure 8.** (a-c): Comparison of the PALM modeled wind speed (WS), resolved TKE (TKE), and SGS-TKE (e) in the place of the ALEGA AIM station for scenarios PALM-ICON (red, base case), PALM-ALAD (green), and PALM-WRF (blue). The gray areas represent night time, white areas the day time.

PALM-WRF at hours 17–18 (see Fig. S19 and Fig. S20 in Supplements). These results confirm findings for the PALM-ICON sensitivity test.

## 4  Discussion

This study confirms that stable conditions represent a challenge for urban boundary layer modeling. In addition to the challenges in modeling meteorological quantities in this situation, the air quality modeling in the emission-ladened street canyon

adds an additional problem that increases the instability of the results. When the amount of the pollution transferred out of the canyon by its ventilation drops near the level of the pollution produced by emission, the pollutants start to accumulate in the



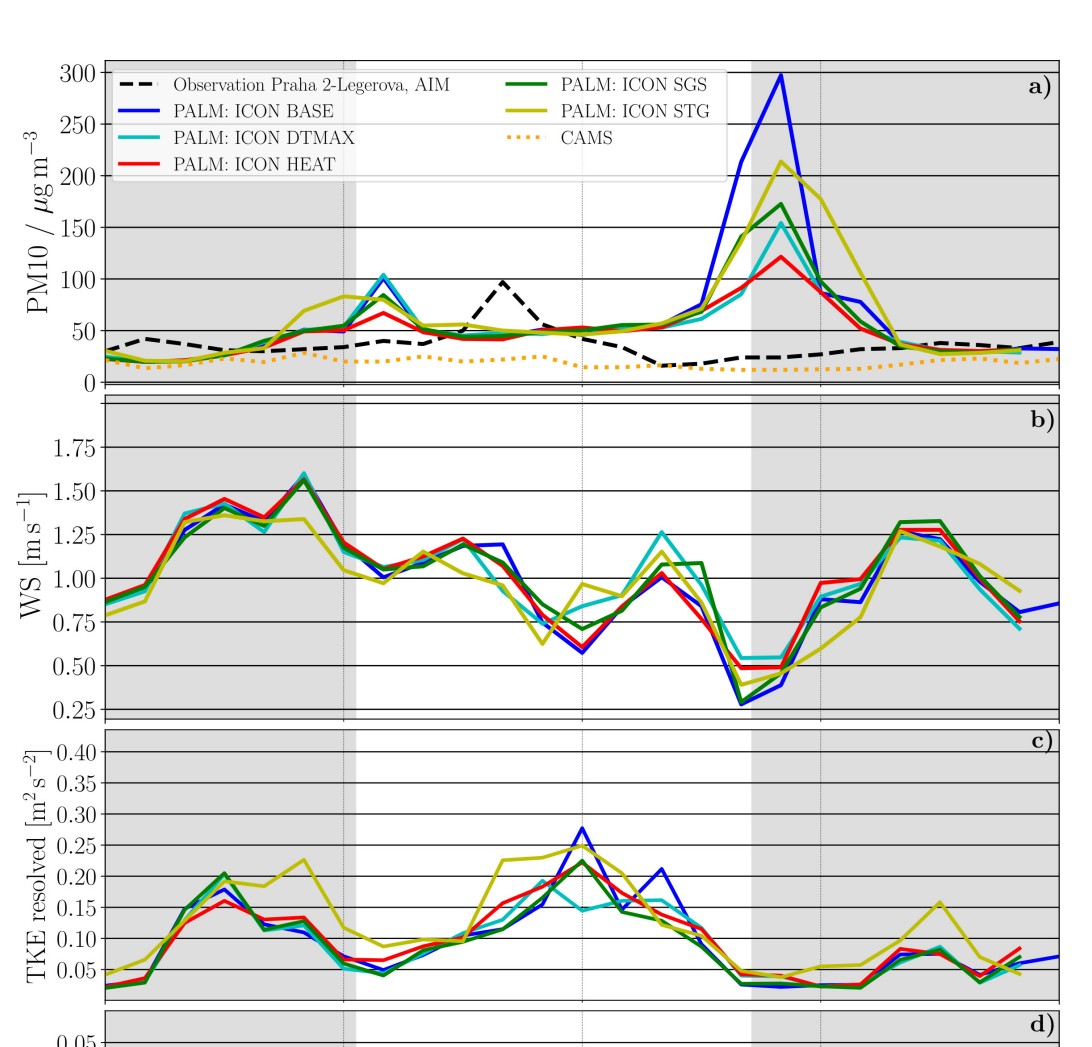

**Figure 9.** (a-d): Comparison of the PALM-ICON modeled values from the scenarios BASE (blue), HEAT (red), SGS (green), STG (yellow), and DTMAX (cyan) on the AIM station ALEGA. The graph (a) shows a comparison of concentrations of $PM_{10}$ with observed values from the ALEGA station (black dashed line) and mesoscale values used as BC (orange dotted line). Next graphs present modeled wind speed (b), resolved TKE (c), and subgrid scale SGS-TKE (d). The gray areas represent night time, white areas the day time.

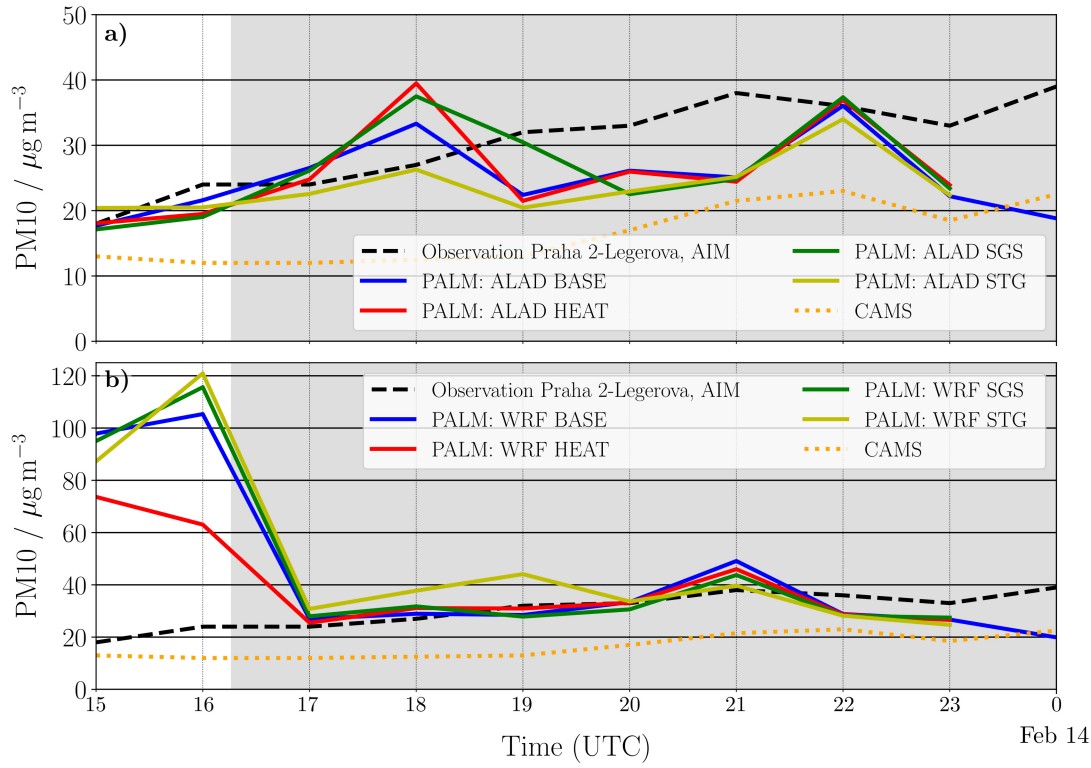

**Figure 10.** (a-b): Comparison of the PALM modeled and observed concentrations of $PM_{10}$ on the AIM station ALEGA. The upper figure (a) shows results for the PALM-ALAD model and the bottom figure (b) for the PALM-WRF model. The graphs show the simulation from the standard configuration of the model (BASE, blue), added anthropogenic heat from cars (HEAT, red), limit of subgrid scale energy (SGS, gren), and adapted profiles for synthetic turbulent generator (SGS, yellow). The gray areas represent night time, white areas the day time.

canyon and each small change in the ventilation can lead to a large change of the pollution concentrations and increase them to unrealistic values.

The comparison of the PALM results driven by different mesoscale meteorological models confirms the crucial role of
the correct meteorological BC for the comparability of the PALM results with observations as previously observed, e.g., by Radović et al. (2023). Fig. 11, which compares concentrations and airflow inside the street canyon for PALM-ICON and PALM-ALADIN, shows how BC taken from two mesoscale models simulating the same real meteorological situation can lead to very different ventilation and air quality inside the street canyon.

An additional reason which contributed to the very low model wind speed can be identified in the boundary conditions
for PALM which were taken from the ICON prediction cycle 15 at 17 UTC and cycle 18 at 18 UTC (see Section 2.4.2) and interpolated in PALM from these two inputs in-between. As can be seen in Fig. 12 showing ICON vertical profiles of



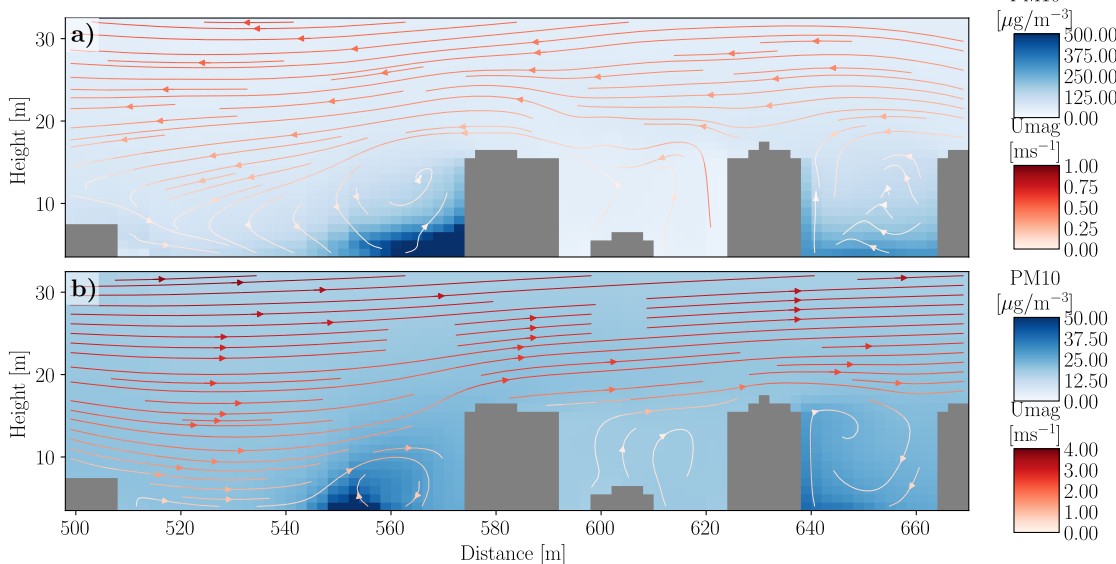

**Figure 11.** (a-b): Comparison of the wind speed and PM$_{10}$ concentrations at the west-east cross-section going through the area of ALEGA AIM station and LIDAR station for ICON (upper) and ALADIN (bottom) scenarios. The values represent the one hour time average from hour 17 to 18. The area of the LIDAR station is placed on the roof of the small building in the left bottom corner of the image, the Legerova street canyon is located between two buildings in the right side of the image. Note that the scales of the figures differ.

the prevailing U-wind component at 17 UTC and 18 UTC, the wind direction is reversed in the cycle 18 prediction. Due to the internal interpolation between boundary conditions representing two opposite wind directions, wind velocities driving the PALM simulation reached values even lower than any ICON predicted value and can be close to zero. This can contribute to
the extreme concentration peak between 17 UTC and 18 UTC through suppressed ventilation.

On the other hand, the strength of the overestimation of concentrations from the PALM-ICON model in hours 16–18, which strongly exceed the values measured in this location throughout the year, suggests that it cannot be fully explained only by the meteorological conditions supplied by the mesoscale models, but it indicates some additional misrepresentation caused by the PALM model itself in the given situation. The sensitivities assess potential of some adaptations to impact this behavior.

The anthropogenic heat from cars causes a significant change in concentrations for the model PALM-ICON only in two time periods (hour 08 and hours 17–18) and for the model PALM-WRF in hours 15–16. It shows that in meteorological situations with larger model wind flow, the additional ventilation induced in the street canyon by this anthropogenic heat is negligible in comparison with the ventilation caused by other turbulent flow. However, during the very stable situation, the

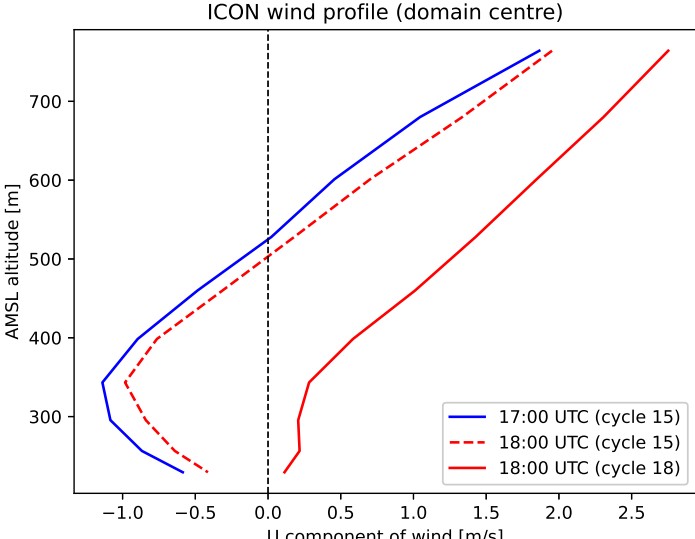

**Figure 12.** : Vertical profile of the U-component of the wind from model ICON at hours 17 and 18 from the ICON prediction cycles 15 and 18 at the PALM domain center.

modeled ventilation can be so low that the additional ventilation induced by this heat can disturb the stability of the air in the

street canyon and that can significantly influence the transport of the pollutants. The figure Fig. 13 compares the modeled air temperature and concentrations of $PM_{10}$ in the west-east cross-section of the PALM child domain in the area of the ALEGA AIM station at the peak hour. The changes of the air temperature caused by the additional heat in the canyon of Legerova street do not exceed 1.2 K but the decrease of the concentrations is substantial about 66%. Moreover, the increase of the temperature and decrease of the concentrations reaches up to the height of 100 m above the surface (see Fig. S21 in Supplements), which

shows that this adaptation also increases the exchange between the street canyon and the layers above the roofs. It suggests that the implementation of this feature in the PALM model would be desirable for air quality modeling in stable meteorological conditions.

The scenario SGS sets a lower limit for SGS-TKE, which controls the modeled diffusion coefficient. That subsequently influences the ventilation of the street canyon and the dissipation of the chemical pollutants. The modeled concentrations only

differ from the base case in times of strong model overestimation, i.e., in hours 07–08 morning and mainly 17–19 evening. Fig. 14 shows comparison of the modeled SGS-TKE in the street canyon and its surroundings for hours 15 (situation before the peak episode) and 18 (peak episode). Before the peak period hour, turbulence is created in the interaction of the airflow with the urban canopy layer and the forced limitation of SGS-TKE is applied only in the upper layers of the domain and has negligible effect inside the street canyon. In the peak hour, the values of SGS-TKE are low in the whole cross-section and the

limit is applied in most parts of the cross-section. This explains the small changes in the concentrations inside the street canyon outside the peak episode.



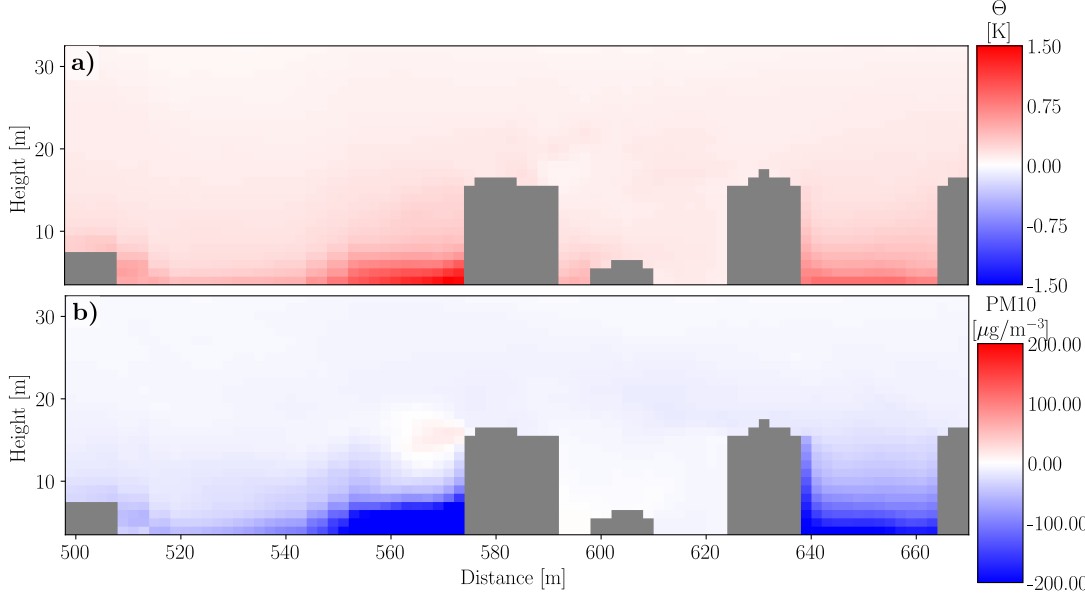

**Figure 13.** (a-b): Difference of the HEAT minus BASE scenarios for the PALM-ICON model on 13 February 2023 at hour 18. The values of potential temperature $\Theta$ (upper) and the concentration of $PM_{10}$ (middle) on the cross-section through the child domain in the position of the ALEGA station are shown. The view is limited to the area from the LIDAR station to Legerova street.

This test shows the potential of improvements in SGS modeling. An approach that would limit the scalar diffusivity could be considered for partial mitigation of the underestimation of the modeled ventilation in the street canyon in stable situations. However, before its real application, more investigation and tests are needed to determine a proper procedure and the limit

values under different meteorological conditions and/or land cover properties for particular vertical levels. Also, the impact of such setting on other modeled values such as the resolved flow and turbulence or the concentrations of $O_3$, NO or $NO_2$, needs to be tested (compare e.g., Makar et al., 2014).

The STG sensitivity test represents another approach where the local street canyon ventilation is not influenced locally, but by changing the global TKE structure induced by the changes of TKE on the parent domain boundary. Fig. 15 shows the

comparison of TKE for BASE and STG scenarios for the parent and child domains at the peak hour. If the dominant length scale (which is parametrized in the model) is below the effective resolution of the advection scheme (which is roughly 80 m in the parent domain), the added perturbation dissipates. That can be partly seen in the results for the parent domain. It indicates a potential of improvements of the profiles of turbulent stresses and length-scales supplied to the STG. The remaining added





**Figure 14.** (a-d): The SGS-TKE for scenarios SGS and BASE on the child domain west-east cross-section through the area of the ALEGA AIM station. The figures show the BASE (a,c) and SGS (b,d) time average of SGS-TKE for the time between hours 13–14 (a,b) and 17–18 (c,d). The view is limited to the area from the LIDAR station to Legerova street.

turbulence inside the domain influences mainly the higher levels while its influence to the low levels directly communicating
with the street canyon is much lower. Despite this, the change of the TKE structure leads to increased street canyon ventilation



and to decrease the $PM_{10}$ concentrations by approximately 30% at hour 18. It shows that potential future adaptations of the STG in PALM can be beneficial specifically for such stable meteorological conditions and that their assessment and testing shall include episodes of this type.

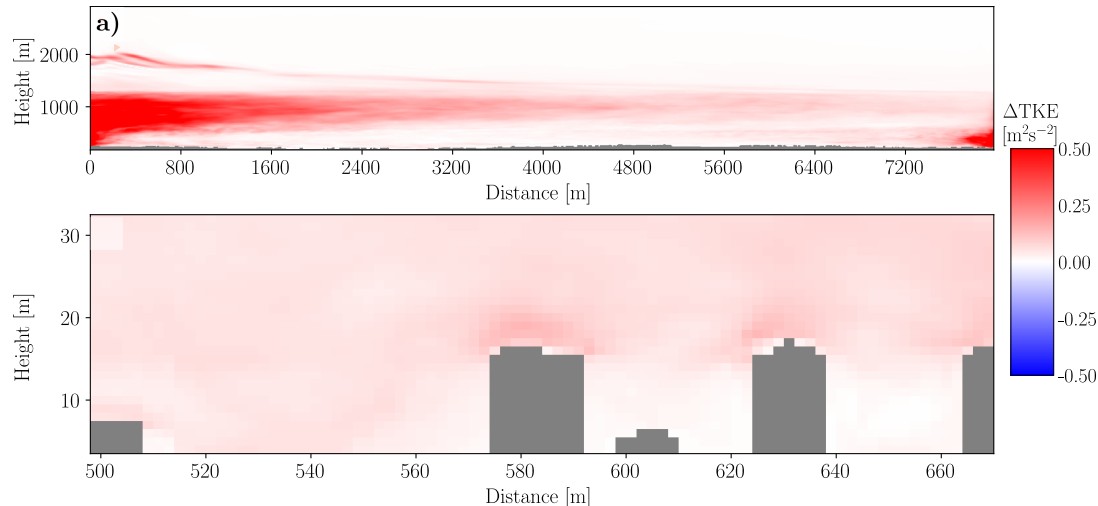

**Figure 15.** (a-b): Differences of the modeled TKE for the scenarios STG and BASE for parent (a) and child (b) domains on the vertical west-east cross-section led in the position of the ALEGA AIM station. The child domain view is limited to the area from the LIDAR station on the left side to Legerova street canyon on the right side. Values represent the time average from hour 17 to 18.

The DTMAX scenario differs from all other scenarios as it does not adjust physical processes in the model but it tests the
possible inaccuracies in the numerical scheme used in the PALM during a stable situation.

Scalar dispersion, especially close to the source location, can be biased due to non-monotone discretizations of the advection term, where numerical errors lead to significant small-scale over/undershots associated with numerical dispersion errors and smoothing of small-scale patterns associated with dissipation errors (Ardeshiri et al., 2020). Even though we employed a flux-limiter scheme after Skamarock et al. (2008) for the vertical advection term to avoid over- and undershooting scalar
concentration, scalar dispersion can still suffer from dissipation errors. Moreover, the advection scheme is only conditionally conservative, i.e. scalar conservation is achieved as long as the flow divergence is negligible. But this is not the case in simulations with terrain e.g. near walls, where the divergence can be significant. Depending on its sign, the advection term can become a source or sink of scalar despite the correction term present in PALM at grid cells with solid wall borders.



To avoid dissipation errors and assure a sufficient source representation by the numerical grid, preliminary measures were taken during preparation of the emission. The strong point sources (like, e.g., power plant chimneys) are rare in the Prague center and the preliminary test simulations proved that the existing ones have negligible impact on street canyon concentrations in the studied area. To avoid potential dispersion error around these sources, they were omitted from the emission inputs. Further, emissions from transport were horizontally dissolved to the full street lines, parking places and other relevant transport areas and emissions from the local heating were dissolved to the whole area of the building roofs.

As both these issues can be mitigated by decreasing the time step, this test highlights the role of discretization errors. The result of the simulation shows that apart from the concentration peak, none of these effects significantly affect the results. In the two peak hours 16–17, this adaptation decreases the concentration by up to 50%, which suggests possible problems with one or both of the discussed issues. To distinguish between them, as well as to test suitable mitigations (e.g. utilization of other advection schemes or testing of the improved divergence correction for scalar transports recently added to PALM), is out of the scope of this study and will require further investigation. But this test shows that in practical simulations of such extremely stable conditions, these effects can take place despite measures taken in preparation of the simulation.

## 5   Conclusions

Stable conditions still represent a challenge in LES modeling, especially for air-quality modeling in poorly ventilated urban canyons where the lack of properly resolved turbulent eddies can lead to unrealistic predictions of air-quality measures. At the same time, stable conditions are very important for the air quality assessment as they are typically a precursor of bad air quality conditions. During the studied stable conditions, PALM underestimates the ventilation in the street canyon which leads to strong overestimation of the air pollutants concentrations. This consequently leads to biased air pollution statistics for the whole episode which could lead to wrong conclusions in practical applications. The substantial cause for the underestimation of the ventilation in the street canyon was the BC supplied by the model ICON at given hours while the same simulation with alternative model ALADIN did not suffer any overestimation of the concentrations. Moreover, at such stable conditions, induced by the ICON-provided BC, the PALM model becomes sensitive to parameters and settings which play no significant role during other conditions. It shows that the assessment of the importance of the particular future model adaptations and development needs to be performed also in such extreme conditions.

Some suggestions for future model development and its applications follow from this study:

- The meteorological conditions used for IBC play a crucial role in the constitution of a system able to produce realistic results comparable with observations. The selection of the most suitable meteorological modeling system providing IBCs should be based on an evaluation performed for every modeled episode. Observations utilized for the evaluation should be located inside the domain and, preferably, the observations directly or indirectly assimilated into this model should not be used for the evaluation. Also, the combination of different mesoscale prediction cycles for creation of BC can harm results in extreme cases.





– For validation purposes, the mesoscale model information, serving as BC for the LES, should be as accurate as possible. For modeling scenarios (e.g., response to changes in urban landcover or emissions or future climatologic scenarios), the boundary conditions do not necessarily need to represent an observed state. However, the represented state should be plausible so as to avoid an unrealistic response to the tested scenarios due to the LES high sensitivity to BC.

– The anthropogenic heat, which was evaluated as a non-significant contributor to the heat in a heat-wave episode in (Juruš et al., 2016) increased modeled air temperature in the street canyon by up to 1.2 K during the studied stable winter conditions. This had a significant influence on the canyon ventilation, causing a 66% decrease of the overestimation of the $PM_{10}$ concentration in the street canyon. Adding the implementation of the anthropogenic heat to PALM is therefore desirable for air quality modeling in stable conditions.

– Limiting of the subgrid-scale TKE is an artificial adaptation which can ensure some basic level of diffusivity in cases where the model does not provide it. The limit $0.02\,\mathrm{m^2s^{-2}}$ used in this study caused a 33% decrease of the overestimation of the concentration during extremely stable conditions. However, a reasonable setting of this limit remains a question. For practical utilization, inhomogeneous setting of this limit based on, e.g., the surface roughness length and vertical level, should be considered. Other adjustments of the SGS parametrization can also be considered, taking into account 520 the large sensitivity of stably stratified turbulence to SGS modeling (see (Couvreux et al., 2020); (Dai et al., 2021)). This topic requires further investigation.

– The simple adaptation of the STG profiles changed TKE in both PALM modeling domains. The added turbulence partly dissipated during transport through the parent domain but it still influenced the ventilation of the street canyon which caused a decrease of the $PM_{10}$ concentrations by approximately 30%. A better utilization of the STG generator in PALM 525 and the inclusion of TKE modeled by the mesoscale model can improve modeling of the TKE in PALM in stable conditions.

– The decrease of the concentrations achieved by limiting the time step during the stable situation suggests issues with dispersion errors and/or mass conservation in the advection scheme used in PALM in such extreme conditions despite employing a flux-limiter scheme and the utilization of the spatially dissolved emissions. This issue can influence the 530 reliability of the results and requires further investigation.

*Code and data availability.* Observations used in this study are available in the repository https://doi.org/10.5281/zenodo.10655033 under the Creative Commons Attribution 4.0 International license. PALM source code and input data used in the simulations are available in the repository https://doi.org/10.5281/zenodo.10998235 under licenses Creative Commons Attribution 4.0 International (data) and GNU General Public License v.3 (source code). The tool PALM_METEO is available under GNU General Public License v.3 from the repository https: 535 //doi.org/10.5281/zenodo.11061001. The emission model FUME is available under GNU General Public License v.3 from the repository https://doi.org/10.5281/zenodo.10142912 and from https://github.com/FUME-dev





## Appendix A

### A1    Climatological description of Prague

In general terms, Prague is typical for a humid continental climate with warm to hot summers (Köppen-Geiger climate type
Dfb). The coldest month is January with a daily mean air temperature of 0.4 °C (1991–2020 climate normal). In recent years,
periods of mild temperatures in winter have been observed, especially in the period between years 2021 and 2024. Summers
in Prague are usually characterized by long sun irradiation; the daily mean air temperature in July, the hottest month, was
20.4 °C (1991–2020 climate normal). Precipitation in Prague is rather low (543.1 mm per year; 1991–2020 climate normal)
since the city is located in the rain shadow of the Krušne hory. The driest season is usually winter (February with a monthly
mean of 21.8 mm) while late spring and summer can bring quite heavy rain, especially in the form of storms (around 80 mm
per month in June, July, and August). Between mid-October and mid-March, temperature inversions are relatively common,
bringing foggy, cold or freezy days. Inversions in the wintertime are conducive to higher air pollution. Prague is also a windy
city with prevailing western winds and an average wind speed of 4.5 ms$^{-1}$ that helps to break temperature inversions and clear
the air in cold months.

*Author contributions.* Jaroslav Resler - leading TURBAN project, PALM development and simulations, emission processing; Petra Bauerová,
Josef Keder, Adriana Šidelářová - observation campaign, observation processing, observation expertise; Michal Belda - data processing, post-
processing, meteorological and air quality expertise; Martin Bureš - data preparation and postprocessing, PALM model development, PALM
simulations; Kryštof Eben - mesoscale simulations; Vladimír Fuka - meteorological expertise, data postprocessing; Jan Geletič - input data
collection and processing, postprocessing; Radek Jareš, Jan Karel - preparation of emission data; Pavel Krč - PALM model development,
PALM simulations, input data processing, postprocessing; William Patiño - postprocessing; Jelena Radović - input data processing and test-
ing, postprocessing; Hynek Řezníček - data preparation and postprocessing, meteorological expertise; Matthias Sühring - meteorological and
PALM expertise; Ondřej Vlček - observation processing, emission processing, air quality boundary data, air quality expertise. All authors
contributed to the experiment design and the manuscript text.

*Competing interests.* The authors declare that they have no conflict of interest.

*Acknowledgements.* This work was supported by project TURBAN (TO01000219; TURBAN – Turbulent-resolving urban modeling of air
quality and thermal comfort) supported by Norway Grants and Technology Agency of the Czech Republic ((TURBAN, 2024a) and (TUR-
BAN, 2024b)). The PALM simulations, and pre- and postprocessing were performed on the HPC infrastructure of the Institute of Computer
Science of the Czech Academy of Sciences (ICS) supported by the long-term strategic development financing of the ICS (RVO:67985807)
and on facilities of the super-computing center IT4I supported by the Ministry of Education, Youth and Sports of the Czech Republic through



the e-INFRA CZ (ID:90254). The co-author M.S. was supported by the Federal German Ministry of Education and Research (BMBF) under

grant 01LP1601 within the framework of Research for Sustainable Development (FONA).



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
