# Peer review of "Challenges of high-fidelity air quality modeling in urban environments - PALM sensitivity study during stable conditions"

_EGUsphere, 2024_

## Referee Comment (RC2)

**Review**

**Title: Challenges of high-fidelity air quality modeling in urban environments – PALM sensitivity study during stable conditions**

In this paper, challenges related to the PALM, especially for air-quality modeling under stable conditions is discussed. Several sensitivity tests (e.g., initial and boundary condition and PALM processes) were performed to investigate the reason for the overestimation of the PALM's air pollutants concentration within urban canyon, which is caused by the underestimation of the estimate of the ventilation. The authors conclude with some suggestions for future model development. The topic is important, and the manuscript is generally well-written and streamlined. The introduction provides a complete (theoretical) background to the study. The scientific merit of the study deserves publication. Yet, I recommend minor revision of the manuscript before its acceptance. This recommendation is based on the comments and remarks listed below:

1) The word '*to*' in line 3 should be changed to "*on*".
2) The word '*unrealistically*' in line 4 should be changed to "*unrealistic*".
3) In line 6, the phrase "… changes of meteorological…" should be changed to ""… changes on meteorological…"
4) Comma is missing in line 20 after the word "radiation" in the phrase "… radiation wind field …"
5) In line 21, change "… a prerequisite for …" to "… the tools needed for …".
6) In line 28, change the phrase "physical bases" to "parameterizations".
The sentence "*However, high-precision and turbulence-resolving methods in the models alone are not sufficient for them to be considered fully reliable for urban atmosphere research, especially in the realm of air quality in the cities.*" is not clear. It should be rewritten.
7) The comma in line 56 after "different scenarios" should be change to full stop.
8) In figure 1, what is the height in which the concentration time series is computed from?
9) Move the flow statistics of the observation and PALM simulations to figure 1 from the Supplements. This will show readers that ventilation in the street canyon is underestimated.

---

## Author Comment (AC1)

**Referee #1**

We would like to thank the reviewer for the careful reading of the manuscript and for the inspiring and helpful review.

*This paper is valuable in that it utilizes the PALM LES model to perform air quality modeling in Prague, Czech Republic under stable conditions, identifying and presenting potential factors that can cause simulation errors. The research is based on various efforts, including elaborated experimental designs and the observation campaign, and the structure is well organized. Based on the manuscript, the following comments are proposed:*

*- Major comments:*

*1) This study focuses on $PM_{10}$ verification at a single point, even though the scale of the experiment and observation sites is quite large. It would be good to first discuss the overall chemical or meteorological field simulation performance of the PALM.*

This experiment is a part of the larger effort to validate the PALM performance in multiple episodes covering different meteorological situations over the whole year against all observations available. However, the aim of this particular experiment was to study one striking phenomenon that needs to be settled before any regular validation can take place. For this reason, the design of this study was intentionally limited to make this study strictly focused on the studied problem.

   The decision to study only the concentration of $PM_{10}$ as a proxy for the street canyon ventilation was given for practical reasons. This selection not only allowed us to treat the species as a passive tracer in given conditions but also strengthened the link between the street level concentrations and street canyon ventilation as most of the non-transportation emissions of $PM_{10}$ are negligible in the studied area. This fact also allowed us to neglect the emission of $PM_{10}$ from the point sources as these sources consist mainly of the chimneys from the burning of the natural gas and their $PM_{10}$ emissions are orders of magnitude lower than transportation emission inside the PALM domains and their impact on street level $PM_{10}$ concentrations is negligible (verified by a testing simulation). This helped us to avoid the strongest source of the discretization errors discussed in the manuscript. In the case of $NO_X$ , the emissions from the point sources and residential heating have a significant impact on the street canyon concentrations and cannot be neglected. We are currently working on the reliable dissolving module for the point source emissions in the FUME emission model to allow us to perform validation of all observed species including influence of the modeled chemistry. In this manuscript, we extended the reasoning of the selection of $PM_{10}$ species for this experiment in section 2.4 to better justify this choice.

   The decision to exclude the full comparison against the sensors from the manuscript itself was done during the process of the manuscript preparation as this additional comparison did not

bring any substantial new information about studied phenomenon to the information provided by comparison with the referential AIM station ALEGA and made the text less focussed and more complicated. Moreover, the sensor observations are less reliable than observations provided by the regulatory AIM stations and their regular discussion would be out of the experiment scope. This spatial analysis can provide interesting new findings when the issue studied in this manuscript partly overriding the information is eliminated. We plan to include such an analysis as part of the currently prepared full validation. Moreover, the basic statistical analysis of the model performance including Taylor diagrams and Q-Q plots is included and discussed in section 3 of our other manuscript Patiño et al.: On the suitability of dispersion models of varying degree of complexity for air quality assessment and urban planning (currently under review, preprint available at https://doi.org/10.2139/ssrn.4822006). To avoid duplication of this text, we included the reference to this manuscript at the end of section 3.1. Nevertheless, we agree with the reviewer that some basic information about the spatial patterns of the meteorological and concentration fields over the domain should be included in the manuscript. We added the maps of the horizontal fields of $PM_{10}$ concentrations and wind speed into the Supplements and added a brief discussion at the end of section 3.1.

*2) How can we discuss the accuracy of the emission data prescribed in the model, and what IC/BC selection criteria can be presented since the accuracy of it will vary from point to point?*

Discussion of the emission accuracy:
The reviewer is right that the emission represents a challenge and a significant source of the inaccuracies in this type of simulation. For this reason, we made efforts to prepare as precise as possible emission inputs. This process is briefly described in the Sect. 2.4.3 but the reviewer is right that the uncertainty is not widely discussed. The reason for this decision was that this experiment focuses on the phenomenon evidently caused by the meteorological conditions and the consequent model behavior and the impact of the possible emission inaccuracy is only limited in comparison with the influence of the insufficient ventilation in the studied hours. The wider discussion of the emission uncertainties is given in section 4.1.2 of our other manuscript Patiño et al.: On the suitability of dispersion models of varying degree of complexity for air quality assessment and urban planning (currently under review, preprint available at https://doi.org/10.2139/ssrn.4822006). To address this comment, we added a brief discussion of emission uncertainties and a reference to the detailed discussion in Patiño et al. to section 2.4.3 of the manuscript.

Discussion of the accuracy of IBC:
This is an interesting and very complex question. Based on our previous experiences (see e.g. cited paper Resler et al. 2021), the concentrations from the mesoscale CTM models suffer from quite large inaccuracies and these inaccuracies can significantly hurt the comparison of the results of the microscale model with observations. To reduce this problem, we decided to combine ground level observations with the CAMS ENSEMBLE modeled vertical profiles in this study. The available AIM background stations were selected for this purpose and the

observations from all selected stations were averaged to reduce the possible bias caused by particular conditions of the individual station. This approach is justified by the following reason. The borders of the parent PALM domain of size 8 x 8 km (see Fig. 2) are located mostly inside the built up city area and this justifies the assumption that the conditions will not differ significantly from each other and from background stations located in similar conditions. Moreover, the provided CAMS ENSEMBLE resolution is 0.1 deg. what corresponds approximately to the grid size 11 x 7 km in the given latitudes so the model itself does not provide any better spatial information. The verification of this approach was done by the comparison of the results of the PALM simulation with blank emission (not shown in the manuscript) with selected background stations and sensors inside the PALM domain which demonstrated a very good agreement.

We added a short information about the reasoning for this treatment to section 2.4.2.

*- Specific comments:*

*1)  P2. L20: "and" is omitted between radiation and wind field.*

Corrected.

*2) P9. L201: The observation site name of "Praha 2-Libus" is written with various names like "Praha C-Libus(P8 of supplements) or Praha 4-Libus(Most of the paper)", causing confusion.*

Corrected and unified across all text.

*3) P13, L298: maximal should be changed to 'minimal' because it is the lowest limit.*

Corrected.

*4) P13, L300: It seems correct that Fig. 8 is cited, not Fig. 7*

Definitely, thank you for noticing it. Corrected.

*5) P21, L415: I couldn't catch up on what the cycle means in the sentence.*

The reviewer is right, we haven't mentioned that ICON D2 has a 3-hours' prediction cycle. We added this fact to the section 2.4.2 about IBC and emphasized this in the formulation at the place mentioned by the reviewer. We also replaced the term "prediction cycle" with the more descriptive formulation "prediction run starting at" in the whole manuscript.

*6) Fig. 4 and 7: Captions should be written consistently because the two figures handled similar information. I don't understand why the observed values were drawn differently in both figures even though it was the same time. Additionally, I don't agree with the statement that the results*

*generally follow the BC profiles in P15, L365.*

The figure captions were fully unified in the revised manuscript. The graphs show different vertical extent corresponding to the height of the parent (Fig. 4) and child (Fig. 7) PALM domains, respectively. The remark about this fact was added to the captions of the figures to notice the reader.

We agree with the reviewer that the statement about PALM following the BC profiles was too generic, we reformulated it to point out some major differences between PALM and its driving model performance in the lowest 250 m.

*7) Fig. 4, 5, 7: It would be nice if time was added not only to the captions but also to each figure*

We tested the possibility of the adding of the time information into the area of the graphs but it interfered in the grid and graph lines. We tried to improve the figure caption instead.

*8) Fig. S01-04 in Supplements: In my opinion, the terrain height plotted in yellow is not that important information in those figures. Instead, it would be better to recognize the map if the ocean was colored in skyblue like the lake.*

We agree with the reviewer but these maps are the official maps produced by the Czech national meteorological service (Czech Hydrometeorological Institute) and we have no possibility to adapt them. We changed the attribution in the figure caption to better highlight the author.

**Referee #2**

We would like to thank the reviewer for the reading of the manuscript, appreciation of the experiment's significance, and for the helpful comments.

*In this paper, challenges related to the PALM, especially for air-quality modeling under stable conditions is discussed. Several sensitivity tests (e.g., initial and boundary condition and PALM processes) were performed to investigate the reason for the overestimation of the PALM's air pollutants concentration within urban canyon, which is caused by the underestimation of the estimate of the ventilation. The authors conclude with some suggestions for future model development. The topic is important, and the manuscript is generally well-written and streamlined. The introduction provides a complete (theoretical) background to the study. The scientific merit of the study deserves publication. Yet, I recommend minor revision of the manuscript before its acceptance. This recommendation is based on the comments and remarks listed below:*

*1) The word 'to' in line 3 should be changed to "on".*

Corrected.

*2) The word 'unrealistically' in line 4 should be changed to "unrealistic".*

Both forms could be used and we still prefer the adverb form. The manuscript will undergo the GMD language copy-editing after typesetting, so they will correct it if needed.

*3) In line 6, the phrase "… changes of meteorological…" should be changed to ""… changes on meteorological…"*

We changed the formulation to "...to changes in meteorological boundary conditions…".

*4) Comma is missing in line 20 after the word "radiation" in the phrase "… radiation wind field …"*

Corrected.

*5) In line 21, change "… a prerequisite for …" to "… the tools needed for …".*

The sentence was reformulated.

*6) In line 28, change the phrase "physical bases" to "parameterizations". The sentence "However, high-precision and turbulence-resolving methods in the models alone are not sufficient for them to be considered fully reliable for urban atmosphere research, especially in the realm of air quality in the cities." is not clear. It should be rewritten.*

We agree with the reviewer that parameterizations are important, but different model types are also important (RANS vs. LES vs. Gaussian etc.). We changed "with different physical bases" to "with varying degrees of complexity" and we reformulated the following paragraph.

*7) The comma in line 56 after "different scenarios" should be changed to full stop.*

Accepted.

*8) In figure 1, what is the height in which the concentration time series is computed from?*

The graph in Fig. 1 shows the concentrations in the height of the AIM station ALEGA. The height of the sensor is 3.5 m above ground and this information was added into the caption of the figure.

*9) Move the flow statistics of the observation and PALM simulations to figure 1 from the Supplements. This will show readers that ventilation in the street canyon is underestimated.*

We are not sure what the reviewer means in this comment. The Supplements contain the flow statistics only in section B, tables S01-S09 and in section C, table S10. These tables show the performance of the models WRF, ICON-D2, and ALADIN (not PALM) against sounding observations over six selected episodes through the year and against LIDAR and Microwave observations through the studied episode. These statistics justify the selection of the ICON-D2 model as the primary source of the IBC for PALM simulations and represent auxiliary information for the reader. We thus leave them in the Supplements.